# Inverse relationship between species competitiveness and intraspecific trait variability may enable species coexistence in experimental seedling communities

Jing Yang [1], Xiya Wang[1], Carlos P. Carmona [2], Xihua Wang [1,3] & Guochun Shen [1,3] ✉

Theory suggests that intraspecific trait variability may promote species coexistence when competitively inferior species have higher intraspecific trait variability than their superior competitors. Here, we provide empirical evidence for this phenomenon in tree seedlings. We evaluated intraspecific variability and plastic response of ten traits in 6750 seedlings of ten species in a three-year greenhouse experiment. While we observed no relationship between intraspecific trait variability and species competitiveness in competition-free homogeneous environments, an inverse relationship emerged under interspecific competition and in spatially heterogeneous environments. We showed that this relationship is driven by the plastic response of the competitively inferior species: Compared to their competitively superior counterparts, they exhibited a greater increase in trait variability, particularly in fine-root traits, in response to competition, environmental heterogeneity and their combination. Our findings contribute to understanding how interspecific competition and intraspecific trait variability together structure plant communities.

There is substantial trait variation among individuals of the same plant species[1]. This intraspecific trait variability (ITV) has great potential to impact species coexistence[2,3] because it can affect the competitiveness of species[4–7] and their long-term population growth rates[8,9]. However, the evidence regarding whether ITV promotes or inhibits species coexistence is mixed[10]. For example, while Begon and Wall[11] reported that ITV can promote coexistence by preventing competitively superior species from driving inferior ones to extinction, Hart et al.[12] showed that ITV can accelerate competitive exclusion by increasing the dominance of superior competitors.

Although the relationship between ITV and species coexistence is intricate, the relative magnitude of ITV between competing species has been proposed to be a key determinant of this relationship[13]. For instance, ITV is expected to enhance coexistence when the competitively inferior species exhibit greater trait variation among its conspecifics than the superior species[10,12]. Such is the case of tree species whose seed quality strongly correlates with their competitive ability at the seedling stage, a high diversity of seed qualities can allow competitively inferior species to produce some highly competitive individuals[13]. Moreover, this variability enables some individuals of the inferior competitors to occupy niche space beyond the niche of the superior competitors. Consequently, when this negative relationship between ITV and species competitiveness occurs, some individuals of the inferior competitors have the same or greater trait value (i.e., seed

[1]Zhejiang Tiantong Forest Ecosystem National Observation and Research Station, School of Ecological and Environmental Science, East China Normal University, Shanghai 200241, China. [2]Institute of Ecology and Earth Sciences, University of Tartu, Tartu, Estonia. [3]Shanghai Institute of Pollution Control and Ecological Security, 1515 North Zhongshan Rd. (No.2), Shanghai 200092, China. ✉e-mail: gcshen@des.ecnu.edu.cn

quality) than do those of the superior competitors[14], thus bolstering the persistence of their populations[11,15]. Although this negative relationship has been widely shown to promote species coexistence in mathematical models[12–14,16–20], empirical evidence of this relationship remains insufficient.

There are conflicting predictions regarding the relative magnitude of ITV between competing species. First, if a competitively superior species occupies a small niche, the competing inferior species may exhibit greater ITV to utilize the remaining unoccupied niches in the environment[21]. For example, while superior competitors may benefit from deep roots to monopolize soil resources, inferior competitors might adapt either by extending their roots deeper[22] or by spreading them laterally to access shallow soil resources[23], resulting in a multipeaked root depth trait distribution with higher ITV (Fig. S1a). Conversely, if the superior competitors already occupies most of the niche space[24,25], the inferior competitors may compress its trait values to maximize its survival in the remaining limited niche space. This may result in a smaller ITV in the inferior competitors. Alternatively, inferior competitors might inherently possess lower trait variability[26,27] (Fig. S1b). Furthermore, if the above differences in resource use arise from inherent distinctions between species' fundamental niches[28], the magnitudes of ITV could diverge among species even in the absence of competition. Thus, any of these scenarios could occur even in competition-free environments.

The relative magnitude of ITV among competing species may change over a short time[26,29]. It is widely recognized that phenotypic plasticity can alter many functional traits[30–32], and the direction and magnitude of these trait changes often vary among conspecifics[28]; thus, a trait's plastic response to competition potentially increases ITV. Consequently, phenotypic plasticity resulting from competitive interactions can amplify ITV, resulting in a complex interplay between ITV and competition. For example, when the roots of neighboring plants come into contact, one species may respond by producing deeper roots, mitigating competition[22]. Alternatively, both species may simultaneously exhibit root elongation, intensifying competition[33]. Moreover, the sensitivity of ITV to competition may differ between competing species. For instance, one study showed that after six months of interspecific competition, the extent of the plastic response of the specific leaf area (SLA) of herbaceous plants in Central Europe increased in parallel to competitive suppression, which was quantified based on plant biomass[6]. This implies that the competitively inferior species experienced greater competition-induced trait changes, leading to a greater amount of competition-induced ITV. Similarly, in an experiment with Mediterranean annual plant species, the competitively inferior species experienced greater changes in trait values when competing with competitively superior species, resulting in reduced competitive hierarchies and favoring coexistence[7]. These findings imply that the impact of competition on ITV might be asymmetrical between competitively superior and inferior species[34], triggering the inverse relationship between species competitiveness and ITV. Despite these examples, little is known about how much competition can change the relative magnitude of ITV among competing species or whether plastic responses to competition could alter the hypothesized negative relationship between competitive abilities and ITVs across different species.

The relative magnitude of ITV among competing species can become highly complex when considering multispecies interactions in heterogeneous abiotic environments[35,36]. Unlike simple pairwise competition, multispecies interactions involve more than two species, thus possibly reducing unoccupied niche space as the total number of species increases[37]. However, heterogeneous abiotic environments can expand the overall available niche space, potentially increasing unoccupied niches[36,38]. The uncertainty surrounding vacant niches in such ecosystems makes it difficult to predict how competitively inferior species might adapt their ITV to access untapped resources and respond to competition. Furthermore, abiotic environments may also affect the magnitude of ITV by either directly filtering out individuals with unfit trait values[20,39] or indirectly adjusting interspecific competition, which in turn alters the extent and direction of plastic responses of traits[40]. Therefore, inferior competitors face more complex abiotic and biotic filters (e.g., different environmental filters and various interspecific competitions) in spatially heterogeneous environments than in homogeneous environments, making it uncertain whether the negative relationship between ITV and species competitiveness remains across spatially heterogeneous environments.

Here, we investigated whether the negative relationship exists between species under competition-free, pairwise competition and multispecies competition conditions in both homogeneous and heterogeneous environments. We conducted a two-phase seedling competition experiment (Fig. 1) and monitored ten key traits in 6,750 seedlings of ten coexisting native tree species (Table 1) over a three-year period. Three different competition scenarios were considered: two-species competition in a spatially homogeneous environment (Phase I), multispecies competition in a homogeneous environment, and multispecies competition in a spatially heterogeneous environment (Phase II). Each scenario included two treatments: competition-free and competition. We quantified the mean competitive ability of the species by comparing total seedling biomass between the competition and competition-free treatments in each scenario. We correspondingly estimated the ITVs for each species in both treatments in each scenario using two methods: multidimensional trait space[41–43] and individual trait variability (Bao's coefficient of variation)[6,7,44,45]. Based on the competitiveness and ITVs of these species, we aimed to address the following questions: (1) Do competitively inferior species have larger ITV than competitively superior species−i.e., negative relationships between ITV and species competitiveness – in environments with homogeneous abiotic conditions and no competition? (2) Does this relationship between ITV and species competitiveness under homogeneous abiotic conditions persist in the presence of pairwise competition? (3) Does this relationship under homogeneous abiotic conditions persist in the presence of multispecies competition? (4) Do the patterns change under heterogeneous abiotic conditions? Our results showed that a negative relationship between ITV and species competitiveness is absent from competition-free homogeneous environments but occurs under pairwise or multispecies competition, as well as in heterogeneous abiotic environments. We found that the emergence of the relationship is driven by a greater increase in ITV in the inferior competitors than in the superior competitors in response to pairwise or multispecies competition and environmental heterogeneity.

## Results

### Absence of the negative relationship in a competition-free environment

Under a competition-free and homogeneous abiotic environment, no significant difference was found in the ITV between the competitively inferior and superior species in the eight pairs of species examined in the Phase I experiment (pairwise Wilcoxon test, $V = 23$, $P = 0.273$; Fig. 2a). At the species level, five out of the eight inferior competitors had significantly greater ITVs than did their superior counterparts (black asterisks, Fig. S2a), while the other three inferior competitors (*Cyclobalanopsis myrsinifolia*, *Quercus chenii* and *Schima superba*) had significantly lower ITVs (red asterisks, Fig. S2a). Similarly, there was no significant linear relationship between the ITV and the competitiveness of species under the competition-free treatment in the homogeneous environment of the Phase II experiment ($P = 0.099$; Fig. 2b and Table S1). Furthermore, visualizations of the reduced-dimensional hypervolume of the traits revealed a more detailed but similar pattern (Figs. S3a–c and S4a–c).

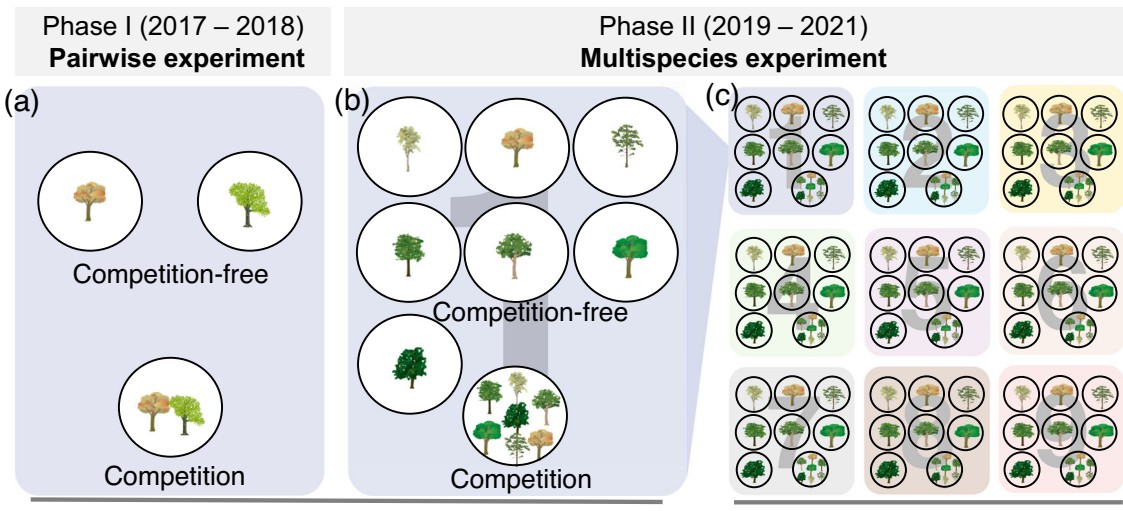

**Fig. 1 | Schematic depiction of the two-phase experimental design.** In Phase I, a pairwise competition experiment (**a**) was conducted in a homogeneous environment with 8 species pairs (Table 2). In Phase II, a multispecies competition experiment was conducted in both homogeneous (**b**) and heterogeneous (**c**) environments with 7 tree species (Table 2). Both phases included two competition treatments: competition-free (one seedling per pot) and competition (two seedlings, one from each competing species in Phase I and seven seedlings, one from each species in Phase II). The heterogeneous environment consisted of nine well-balanced and representative abiotic environmental blocks selected using an orthogonal design approach with three levels (low, medium, and high) for each of the three key factors (light, soil moisture and phosphorus content). Different background colors signify distinct abiotic environments. Further details of the Phase II experiment are depicted in Fig. S12. The detailed environmental data for each environmental block are given in Table S8. The visual elements used in this figure are provided by the Integration and Application Network (IAN, ian.umces-s.edu) at the University of Maryland Center for Environmental Science (UMCES), which are available under the Creative Commons Attribution-ShareAlike 4.0 International (CC BY-SA 4.0) license (https://creativecommons.org/licenses/by-sa/4.0/).

## Emergence of the negative relationship under competition

At the end of our experiment, interspecific competition caused significant reductions in biomass in all the species (Table 2, S2). Visualization of hypervolume showed that competition altered the shape, range, and center of the multidimensional trait hypervolumes of most species (Fig. S4). The three inferior competitors with lower ITVs in the competition-free environment of Phase I had significantly greater trait variability under the competition treatment (Fig. S2b). Consequently, inferior competitors exhibited significantly greater ITV than their superior counterparts in pairwise competition in a homogeneous environment ($V = 36$, $P = 0.004$; Fig. 2d). There was also a significant negative relationship between ITV and the competitiveness of species under multispecies competition in the same abiotic environment (that is, environmental block 1) ($R^2 = 0.60$, slope $= -8.84$, $P = 0.025$; Fig. 2e and Table S1). Furthermore, similar negative patterns were also found in homogeneous environments by mixed-effects models based on 7-dimensional (pseudo $R^2 = 0.061$, slope $= -3.68$, $P = 0.033$; Fig. S5b) and reduced-dimensional hypervolumes (pseudo $R^2 = 0.074$, slope $= -1.55$, $P = 0.017$; Fig. S5d). Similar trends were also found when the two deciduous species were excluded (Fig. S6). All of these results support that an inverse relationship between species competitiveness and ITV occurs under interspecific competition.

We further discovered that the emergence of the negative relationship when transitioning from competition-free to competition treatment was caused by the different responses of competitively inferior and superior species to competition. Inferior competitors exhibited greater increases in ITV than did their superior counterparts in pairwise competition ($V = 35$, $P = 0.007$; Tables S3). Under multispecies competition, the change in ITV when transitioning from the competition-free to the competitive treatment was significantly and negatively correlated with species competitiveness (Table S4). Thus, species that suffered more competitive suppression responded to competition with a greater increase in ITV ($R^2 = 0.61$, slope $= -13.25$, $P < 0.001$; Fig. 3a).

The negative relationship is also influenced by heterogeneous environmental conditions, including light, soil moisture and phosphorus content. In the competition-free treatment, we observed the emergence of the negative relationship with transitioning from homogeneous to heterogeneous environments (Fig. 2b, c). Under competition, the negative relationships between ITV and species competitiveness became even stronger, with changes in slope and $R^2$ from $-8.84$ and 0.60 in homogeneous environments to $-14.96$ and 0.85 in heterogeneous environments, respectively (Fig. 2e, f and Table S1). Most of the change in slope was attributed to the greater increase in ITV for competitively inferior species in heterogeneous environments than for superior species (Fig. 3b).

## Similar patterns at the individual trait level

The patterns of intraspecific variability in individual traits were similar to those observed when considering all traits using multidimensional hypervolume, although there were some differences (Fig. S7). Overall, competitively inferior species had significantly greater intraspecific variability in individual traits than competitively superior species in the competition treatment ($P < 0.001$, black solid line in Fig. S7b), while no significant difference was found in the competition-free treatment ($P = 0.062$, black dashed line Fig. S7a). Specifically, inferior competitors had significantly greater conspecific variability in terms of root tissue density and specific root length in the competitive treatment (Fig. S7b) but lower variability in leaf toughness in both the competition-free and competitive treatments (Fig. S7a, b). In addition, intraspecific variability increased for most of the competing species for all the root and leaf traits that were measured except for leaf toughness, whose variability decreased (Fig. 4 and Table S5). Additionally, three

**Table 1 | Tree species and their growth forms in our two-phase seedling experiment**

| Phase | Latin name | Abbr.[a] | Genus | Family | Growth form |
|---|---|---|---|---|---|
| Phase I/II | *Quercus chenii* | QC | Quercus | Fagaceae | DC |
| Phase I/II | *Hovenia acerba* | HA | Hovenia | Rhamnaceae | DC |
| Phase I/II | *Castanopsis sclerophylla* | CS | Castanopsis | Fagaceae | EG |
| Phase I/II | *Schima superba* | SS | Schima | Theaceae | EG |
| Phase I/II | *Cyclobalanopsis glauca* | CG | Cyclobalanopsis | Fagaceae | EG |
| Phase I | *Lithocarpus harlandii* | LH | Lithocarpus | Fagaceae | EG |
| Phase I | *Cyclobalanopsis myrsinifolia* | CM | Cyclobalanopsis | Fagaceae | EG |
| Phase I | *Phoebe sheareri* | PS | Phoebe | Lauraceae | EG |
| Phase II | *Daphniphyllum oldhami* | DO | Daphniphyllum | Daphniphyllaceae | EG |
| Phase II | *Lithocarpus glaber* | LG | Lithocarpus | Fagaceae | EG |

Phase I: the pairwise competition experiment in a homogeneous environment (Fig. 1a); Phase II: the multispecies competition experiment in the homogeneous (Fig. 1b) and heterogeneous (Fig. 1c) environments.
*DC* deciduous, *EG* evergreen
[a]Abbreviation of species name.

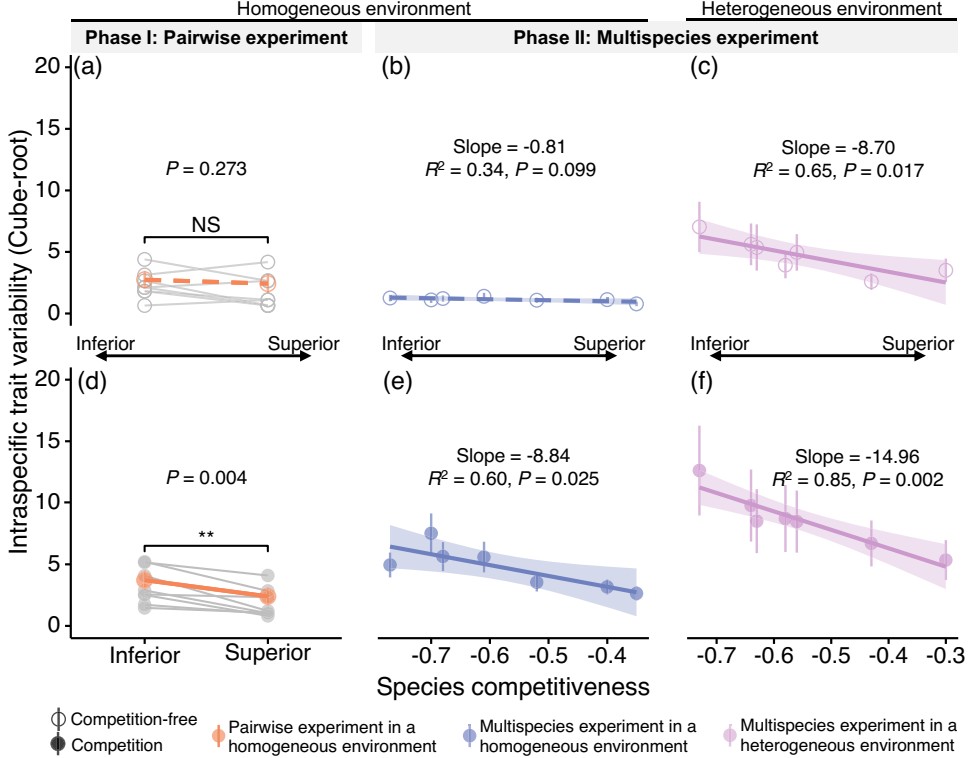

**Fig. 2 | Relationships between intraspecific trait variability (ITV) and competitiveness of species. a, d**, Pairwise experiments in homogeneous environments (*n* = 345 and 661 seedlings, respectively). **b, e**, Multispecies experiments in homogeneous environments (*n* = 132 and 121 seedlings, respectively). **c, f**, Multispeices experiments in heterogeneous environments (*n* = 1005 and 916 seedlings, respectively). ITV per species was quantified by the 999 simulated hypervolume of seven traits (measured in SD[7] units), and competitiveness was quantified using the relative interaction intensity index (*RII*) based on whole seedlings' biomass. Gray (**a, d**), blue (**b, e**), and purple (**c, f**) points with error bars represent the mean and standard error of the 999 simulated hypervolumes for each competing species, transformed by cube root. Hollow points and solid points represent species under competition-free and competition treatments, respectively. Detailed sample sizes for each competing species in two-phase experiment are shown in Table S4. The orange dots in a and d represent the mean ITVs of the eight superior (or inferior) species. The gray and orange line connecting these dots visually depicts the difference in ITV between competing species, and significance was tested by one-sided paired Wilcoxon rank-sum test. Black asterisks indicate levels of significance (•: <0.1; *: <0.05; **: <0.01; ***: <0.001). NS indicates nonsignificant. In **b, c** and **d–f**, solid (*P* < 0.05) and dashed (*P* > 0.05) lines are simple linear regression lines with 95% confidence intervals.

fine-root traits exhibited the greatest relative increase in intraspecific variability (124.80 ± 19.51%, Fig. 4).

## Discussion

Since Begon & Wall[11] first discovered that intraspecific variability in competitively inferior species is important for species coexistence, a growing number of theoretical studies have shown that the negative relationship between ITV and species competitiveness, also known as the trait mean-variance trade-off can promote species coexistence[12–14,16–20]. However, it remains unclear whether and when such a negative relationship exists in real plant communities. To our knowledge, our three-year seedling experiment provides the first

**Table 2 | Competitive rankings between paired species in the Phase I experiment and among multiple species in the Phase II experiment based on relative interaction intensity (*RII*, mean ± standard error)**

| Phase | Pair code | Species | *RII* | Species | *RII* |
|---|---|---|---|---|---|
| | | Inferior | | Superior | |
| Phase I | CM-QC | CM | −0.36 ± 0.02*** | QC | −0.18 ± 0.02*** |
| | CM-CG | CM | −0.25 ± 0.03*** | CG | −0.20 ± 0.02*** |
| | CS-CG | CS | −0.24 ± 0.04*** | CG | −0.23 ± 0.03*** |
| | PS-HA | PS | −0.38 ± 0.02*** | HA | −0.09 ± 0.02*** |
| | QC-HA | QC | −0.31 ± 0.04*** | HA | −0.11 ± 0.03*** |
| | SS-LH | SS | −0.40 ± 0.04*** | LH | −0.22 ± 0.03*** |
| | PS-QC | PS | −0.42 ± 0.02*** | QC | −0.16 ± 0.03*** |
| | CG-QC | CG | −0.36 ± 0.04*** | QC | 0.002 ± 0.03 |
| | | Homogeneous environment | | Heterogeneous environment | |
| Phase II | — | DO | −0.77 ± 0.02*** | DO | −0.73 ± 0.02*** |
| | — | LG | −0.70 ± 0.03*** | LG | −0.64 ± 0.02*** |
| | — | CS | −0.68 ± 0.03*** | CS | −0.63 ± 0.01*** |
| | — | CG | −0.61 ± 0.03*** | CG | −0.58 ± 0.02*** |
| | — | HA | −0.52 ± 0.05*** | SS | −0.56 ± 0.02*** |
| | — | SS | −0.40 ± 0.04*** | HA | −0.43 ± 0.02*** |
| | — | QC | −0.35 ± 0.04*** | QC | −0.30 ± 0.02*** |

The *RII* measures the impact of the species (Phase I) or the other six species (Phase II) on the whole biomass of the focal species. Species with lower *RIIs* experience greater competitive suppression and have lower competitive ability, making them inferior in terms of competitiveness. Notably, species may have different competitive rankings in different competitive environments. Significance of *RIIs* was tested by a two-sided Wilcoxon rank-sum test.
Asterisks indicate that the species, compared to those in the competition-free treatment, were subjected to significant biomass suppression due to competition (***: <0.001), with sample size and exact *P*-value presented in Table S2.

experimental evidence for the inverse relationship in the tree seedling stage, as well as the conditions under which this occurs.

Under a competition-free homogeneous environment, our observations did not support the hypothesis that species with lower average competitive abilities exhibit higher levels of ITV, countering the assumption of a negative relationship between species competitiveness and ITV. This finding is consistent with the trait optimization hypothesis[46], which assumes that plant traits are the outcome, to some degree, of environmental filtering that maximizes the performance of the plants in a given habitat. In a competition-free homogeneous environment, all individuals, regardless of the status of the species in the competitive hierarchy, are subject to similar abiotic selection and use the same set of resources (e.g., Fig. S8). This results in similar variation in most traits between competitively inferior and superior species and lower levels of ITV in a competition-free homogeneous environment than in heterogeneous or competitive environments[47]. In agreement with this expectation, our experiment showed that the ITV of most species (Figs. 2a, b and S3a, b) and interspecific trait dissimilarities among 62.5% of the species pairs (Table S6) were lower in competition-free homogeneous environments than in heterogeneous or competitive environments. In addition to trait optimization, the low genetic variation in conspecific seedlings from a single mother tree in our study may have limited trait variation. Future research should aim to optimize seed collection methods, for example, by selecting parent trees from different habitats in the study area, increasing the representation of genetic variation within species and assessing the ITV of species more accurately.

Under competition, unlike in the competition-free treatment, the ITV of the competitively inferior species increased more than that of the competitively superior species (Fig. 3), which is attributable to the negative relationship between species ITV and competitiveness. This relationship could be because the inferior competitors seem to show greater trait plasticity than superior competitors under competition. In agreement with these findings, in a six-month experiment with annual species, Carmona et al.[7] found that the magnitude of height plasticity in annual species changed proportionally with the intensity of competition (*RII*), as evidenced by a relative decrease in overall biomass. This implies that competitively inferior species experiencing stronger competitive suppression undergo greater competition-induced trait changes. Longuetaud et al.[31] also showed that inferior *Quercus* species had higher crown plasticity, while superior *Fagus sylvatica* exhibited lower plasticity in Western European mixed forests. Consequently, ITV can increase under competition through high plastic trait responses. For instance, Gruntman et al.[48] found that the clonal plant *Potentilla reptans* displayed plastic responses such as confrontational vertical growth, shade tolerance, and lateral avoidance when competing with neighbors of varying heights and densities. These plastic responses often vary substantially in direction and magnitude among conspecifics[28], potentially increasing the ITV of the clonal plant. Indeed, Bittebiere et al.[47] found that competition led to greater ITV in ramet and connected traits of two clonal plants. Given these observations, it is crucial to extend the validation of these findings to a wider range of contexts, as the current inferences are predominantly based on our experimental studies.

Notably, the plastic trait responses under competition do not necessarily mean that competitively superior species will always increase their ITV under competition. The traits of superior species may sometimes converge toward an optimal value through plasticity, resulting in lower ITVs under competition. For example, taller plants often adopt a rapid vertical elongation strategy to maximize light access above the canopy[48] and usually become superior at the seedling stage. This directional change can result in a lower stem-specific density (SSD) for superior competitors, thus reducing SSD variability. In agreement with these findings, we observed that two of the eight competitively superior species in our experiment exhibited reductions in SSD variability (Fig. S9a, d). Moreover, we also observed that four of the eight competitively superior species experienced reductions in multidimensional ITV from the competition-free to the pairwise competition experiments (Table S3). Interestingly, all the superior competitors presented increased ITVs under multispecies competition (square and triangle points in Fig. 3), possibly because most of the species were superior to some but inferior to other species under multispecies competition. Moreover, resource availability can also modulate the association between ITV and competition. For instance, in environments with abundant resources (e.g., environment block 1 with abundant light and water), inferior competitors exhibited higher ITVs (slope of −8.84 in Fig. 2e and S5b), while in resource-constrained conditions (e.g., block 4 with scarce light and nutrients), this relationship weakened (slope of −2.58 in Fig. S5b). This finding aligns with the stress gradient hypothesis[36], whereby species in resource-limited environments experience restricted growth and dampened competition. These findings highlight the need for further exploration of the effect of resource availability on the relationship between ITV and competition.

Our results showed that the inverse relationship between species competitiveness and ITV persists in heterogeneous abiotic environments. In agreement with previous findings[30,49–52], our results showed that ITV not only responds to abiotic environments but also that the response is divergent among competing species. We provided clear experimental evidence showing that competitively inferior species have a greater increase in ITV than superior species in competition-free heterogeneous environments (Fig. 3b), supporting the hypothesis that

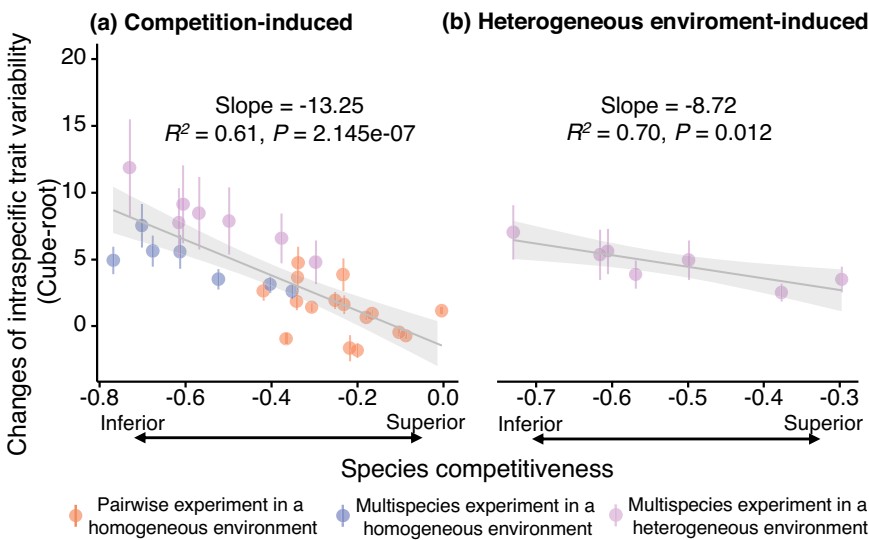

**Fig. 3 | Relationships between mean changes in intraspecific trait variability (ITV) and competitiveness of species. a** depicts mean changes in ITV from the competition-free to competition treatments across three scenarios: pairwise in a homogeneous environment (orange points, $n = 1006$ seedlings), multi-species in a homogeneous environment (blue points, $n = 253$ seedlings), and multi-species in a heterogeneous environment (purple points, $n = 1921$ seedlings). **b** illustrates mean changes from the competition-free homogeneous environment to the competition-free heterogeneous environment. Different colors represent three distinct competition scenarios ($n = 1005$ seedlings). ITV per species was quantified by the 999 simulated hypervolume of seven traits (measured in SD[7] units), and competitiveness was quantified using the relative interaction intensity index (*RII*) based on whole seedlings' biomass. Points with error bars represent the mean and standard error of the 999 simulated hypervolumes for each competing species, transformed by cube root. Detailed sample size for each competing species in two-phase experiment are shown in Table S4. The solid ($P < 0.05$) lines are simple linear regression lines with 95% confidence intervals.

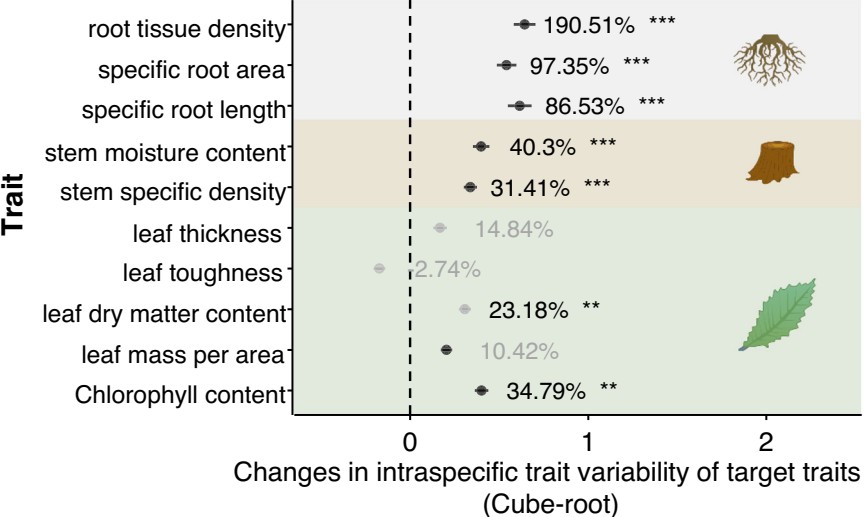

**Fig. 4 | Mean changes in intraspecific trait variability of root, stem and leaf traits from competition-free to competition across three competitive scenarios: homogeneous two-species competition, homogeneous multi-species competition, and heterogeneous multi-species competition.** A positive change (mean ± standard error, $n = 30$ competing species) indicates an increase in trait variability among conspecifics under competition, while a negative value indicates decreased variability. The numbers next to the data points are the relative changes in intraspecific trait variability from the competition-free to competition treatments. Background colors represent the traits of the different organs of the plant (roots, stems, leaves). Black asterisks indicate levels of significance as determined by a two-sided Wilcoxon rank-sum test (•: <0.1; *: <0.05; **: <0.01; ***: <0.001), with the exact *P*-value was shown in Table S5. The visual elements used in this figure are provided by the Integration and Application Network (IAN, ian.umces.edu) at the University of Maryland Center for Environmental Science (UMCES), which are available under the Creative Commons Attribution-ShareAlike 4.0 International (CC BY-SA 4.0) license (https://creativecommons.org/licenses/by-sa/4.0/).

inferior competitors have a broader niche width than superior competitors[53]. The broad niche of these plants enables inferior competitors to alter their traits to adapt to a more diverse range of environments[54], thereby increasing their ITV and survival opportunities[55]. Moreover, interspecific competition can further drive some individuals of inferior competitors to adjust their trait values to exploit resources that superior competitors cannot efficiently utilize, leading to an even greater increase in ITV under interspecific competition in a heterogeneous environment (Fig. S10). Our results indicated the largest increase in ITV, and the strongest negative relationship occurred in the competition treatment in a heterogeneous environment (Fig. 2f).

The different responses in the ITV between competitively inferior and superior species led to two important insights. First, they emphasize that both intraspecific and interspecific trait variations are not constant but can change in response to competition and abiotic

conditions[6,47]. These responses can be substantial, exceeding two orders of magnitude (e.g., competition induced an average 23.77-fold increase in ITV in our experiments), and cannot be ignored when predicting population and community dynamics based on trait differences[7,27,56]. Second, although these responses are complex, there is a consistent trend where competitively inferior species exhibit greater increases in ITV under competitive or heterogeneous environments. This trend has been ignored in previous competition models, but our results demonstrate that it is important to incorporate this trend in such models to understand species coexistence and even the mechanisms of species invasion. For instance, invasive species exhibit greater phenotypic plasticity than native species[57], which could allow invasive species to survive the competition of native species. However, to fully understand the effect of the trend on coexistence, more comprehensive experiments, such as response surface competition designs[58] that incorporate contemporary species coexistence theory, are needed.

Our results showed that fine-root traits had significantly greater changes in ITV (e.g., +190.51% in root tissue density) than leaf and stem traits (Figs. 4, S7c, d and Table S5). There are two possible reasons for the observed changes. First, in our study, seedlings may have experienced greater belowground competition than aboveground competition[59]. Since root traits directly influence belowground resource acquisition[60,61], plants may exhibit greater plasticity in response to belowground competition for space, water, and nutrients[62,63]. Strong competition could drive large changes in the root trait ITV[6]. In addition, two of the three abiotic variables (soil moisture and phosphorus levels) manipulated in our experiments were belowground factors. Changing these soil variables may increase root variability, which is consistent with the findings of previous studies[64,65]. Since leaf and stem traits are not directly influenced by belowground resources, changes in their ITV are expected to be smaller.

In contrast to the overall ITV pattern, leaf toughness (LTO) exhibited the opposite trend—competitively inferior species had significantly lower LTO variability than superior species (yellow–brown lines in Fig. S7a, b). Furthermore, competition tended to decrease LTO variability (Figs. 4 and S7c). Two key factors may have driven this result. First, there might be internal resource usage trade-offs among traits since plastic trait responses consume nutrients, and the total stored nutrients are finite. Expanding variability in many important traits may first restrict variability in other traits[66]. Second, our greenhouse plants were protected by insect screens, and all seedlings were regularly sprayed with pesticides to minimize the effects of herbivory. This likely reduced the benefits of high LTO variability in our experiment. Overall, resource limitation and low herbivory pressure could have jointly contributed to the opposite LTO variability observed in our study. Overall, different traits can respond differently to competition during their ITV.

Two major limitations of our study are worth noting. First, while we have shown a strong negative relationship between species competitiveness and ITV, empirical tests of its importance for species coexistence are still lacking[67]. Future work considering long-term population dynamics, sophisticated experimental designs such as response surfaces[68], or a combined experimental–theoretical approach[9] could address this problem. Second, our experimental duration of three years was relatively short compared to the life cycle of the trees. Therefore, it is unknown whether the negative relationship observed in our seedling stage may also apply to natural forests.

In summary, in our seedling experiment, we investigated the negative relationship between species competitiveness and ITV in both pairwise and multispecies competition, as well as in homogeneous and heterogeneous environments. The negative relationship was not observed in a competition-free homogeneous environment, but it emerged in interspecific competition or heterogeneous environments. The emergence of this negative relationship was attributed to the different responses of competing species to competition or heterogeneous environments, specifically to inferior competitors having a greater increase in intraspecific variability than superior competitors. To our knowledge, these findings provide the first experimental evidence for this negative relationship, providing empirical evidence for its potential importance in species coexistence. The distinct responses of inferior and superior competitors not only explain the emergence of inverse relationship but also offer new avenues for understanding the effects of intraspecific variability on species coexistence.

## Methods
### Study site and species selection
The experiment was performed in a greenhouse located at the southern foot of Taibai Mountain in Tiantong National Forest Park, Ningbo, Zhejiang Province, China (29° 48.817′ N, 121° 47.116′ E), which has a subtropical monsoon climate with hot and humid summers, dry and cold winters, an annual mean temperature of 16.2 °C and precipitation of 1374.7 mm. The soil in the study area is a montane yellow–red soil characterized by a predominantly loamy and clay texture with a slightly acidic soil pH ranging from 4.4 to 5.1. The forests in this area are subtropical evergreen broad-leaved forests with deciduous tree species[69].

We initially selected 16 dominant and codominant native tree species based on previous knowledge about the forest community composition in the area[69], life forms (11 evergreen and 5 deciduous species, as listed in Table S7), phylogenetic relationships (Fig. S11), and seed availability. In 2016 and 2018, more than 2000 healthy and intact seeds per species were collected. Specifically, we exclusively sourced seeds from a single mature, healthy adult tree located within our study area. This approach was used to minimize the genetic variation among conspecific seeds and maternal effects among species.

These collected seeds were then sterilized with insecticides (imidacloprid and carbendazim) before being stored over winter for sowing in seedling trays in April of the respective experimental years. Throughout the germination period, the seedlings were exposed to full sunlight, and the seedlings were adequately watered. The soil used for seedling rearing was collected from the topsoil (15–30 cm, excluding nitrogen-rich soil at 0–15 cm depth) of the forest where the mother trees grew. The soils were sieved through a 1-cm mesh to remove large particles and organic debris and homogenized before the experiment. The prepared soil had 6.10 g/kg total carbon, 0.4 g/kg total nitrogen, 0.45 g/kg total phosphorus and a pH of 5.21. Due to the low germination rates of some species (Table S7), we ultimately reduced the number of experimental species to 10, with 8 species in Phase I and 7 species in Phase II (Table 1). The seedling height at transplantation was approximately 6–25 cm, with deciduous species averaging 12–25 cm and evergreen species averaging 6–13 cm (Table S7). This difference in height between deciduous and evergreen species may cause interspecific asymmetric competition, which is observed at the seedling stage in tree species[70].

### Experimental design
To test the assumption that a negative relationship exists between species competitiveness and ITV across various competition scenarios, we conducted a two-phase tree seedling competition experiment over three years (Fig. 1). In Phase I (2017–2018), we examined the relationship in a two-species pairwise competition in a homogeneous environment (Fig. 1a). We expanded this examination to Phase II (2019–2021) in which multispecies competition in both homogeneous (Fig. 1b) and heterogeneous abiotic environments (Fig. 1c) were considered.

**Phase I: pairwise competition in a homogeneous environment.**
From June 2017 to August 2018, we carried out a pairwise competition test with eight tree species in a spatially homogeneous environment

within a greenhouse. Given the constraints of seedling availability, we chose eight species pairs in this phase (Table 2). The species pairs were selected to have an even distribution of phylogenetic distances among species (Fig. S11) to maximize the representation of species with different competitive abilities based on the competition-relatedness hypothesis[71]. In June 2017, we transplanted seedlings of the eight selected species into pots with a diameter of 15 cm and a height of 15 cm. The seedlings were divided into two planting treatments (Fig. 1a): competition-free and competition. In the competition-free treatment, only one seedling of each selected species was planted in a single pot. In the competition treatment, two seedlings, one from each competing species, were planted in a pot. To ensure a fair comparison between treatments, the initial height and base diameter of the seedlings were kept similar for each species among the pots. If a seedling died within the first month after transplanting, it was replaced with a similar-sized conspecific seedling. We carefully kept the distance between two seedlings within a pot within 3–5 cm to facilitate competition among seedlings within the pots while keeping the interpot distance fixed at 15–20 cm to minimize potential interference between pots. For each species in the competition-free treatment and for each species pair in the competition treatment, 45 replicate pots were established, resulting in a total of 720 pots with 1080 seedlings. A total of 78 individuals with at least one missing trait value were excluded from the following analyses in Phase I.

Since we focused primarily on the effects of competition on ITV, we installed insect screens and regularly sprayed pesticides to prevent potential herbivory. We also maintained similar abiotic conditions (particularly light intensity, soil moisture, and soil nutrient content) within and between all pots in Phase I. To ensure this homogeneity, all pots contained the same quantity ($2 \pm 0.01$ kg) of well-mixed soil as that used for germinating seedlings. Additionally, the light intensity was kept as similar as possible in the greenhouse, and the pots were randomly rearranged every two weeks during the experiment to minimize the effects of ambient conditions. The soil moisture was maintained at approximately $33.04 \pm 0.28\%$ by adding 200 ml of water to each pot every 3–5 days. In this study, we refer to the experiment in Phase I as a two-species competition experiment in a homogeneous environment.

**Phase II: Multispecies competition in homogeneous and heterogeneous environments.** In June 2019, we conducted a multispecies competition experiment with seven tree species (Table 1) in nine abiotic environmental blocks (Fig. 1c) in the same greenhouse. Seedlings within each environmental block shared the same abiotic conditions, so that each environmental block represented a spatially homogeneous environment. However, seedlings in different environmental blocks experienced different abiotic conditions; thus, the nine environmental blocks comprised a spatially heterogeneous environment (Fig. S12), similar to the findings of the studies of Allesina & Levine[72]. Both the homogeneous and heterogeneous environments included competition-free and multispecies treatments.

The nine abiotic environmental blocks (Fig. 1c) were selected according to a three-factor and three-level orthogonal design (Table S8). These abiotic environments effectively represented the abiotic conditions in the study area while considering practical constraints and experimental feasibility. The three factors considered were light (L), soil moisture (M), and soil phosphorus content (P), all of which are critical for seedling growth. Different layers of shade netting around and on top of the greenhouse, along with varying watering frequencies and fertilizer concentrations (see next paragraph for details), were used to control the three abiotic environmental factors at low, medium and high levels (represented by 1-3) based on real environmental data from previous surveys[69]. The abiotic conditions for the nine blocks (named blocks 1-9) were as follows: L3-M3-P1, L3-

M1-P3, L3-M2-P2, L2-M2-P1, L1-M1-P2, L1-M3-P3, L2-M2-P3, L2-M1-P2 and L2-M3-P1 (Fig. S12). In Block 1 (L3-M3-P1), the abiotic environment was identical to that in Phase I, which facilitated a direct ITV comparison between pairwise and multispecies competition.

Specifically, two-layer and one-layer shade nets around and above the plants created two light gradients: L1 and L2, respectively. The third light level, L3, was one with no shade net. The mean ($\pm$SE) light intensity, measured by an LI-1500 (LI-COR, USA) on a sunny summer day from 8 am–5 pm, was $14.54 \pm 1.99$ µmol·m$^{-2}$·s$^{-1}$ in L1, $502.91 \pm 103.09$ µmol·m$^{-2}$·s$^{-1}$ in L2, and $1237.20 \pm 195.45$ µmol·m$^{-2}$·s$^{-1}$ in L3. For soil moisture, watering intervals of six days (M1), four days (M2), and two days (M3) were utilized in summer, reducing these intervals by one day for every 3 °C temperature decrease. The mean ($\pm$ SE) volumetric soil moistures throughout the year were $21.36 \pm 0.28\%$ in M1, $28.76 \pm 0.22\%$ in M2 and $33.04 \pm 0.28\%$ in M3. To create three levels of soil phosphate availability, each pot was fertilized monthly with 200 ml of 0 g·ml$^{-1}$ (P1), 0.2 g·ml$^{-1}$ (P2) or 0.4 g·ml$^{-1}$ (P3) liquid phosphate (NaH$_2$PO$_4$). After two years, the mean ($\pm$ SE) soil phosphate concentration, measured by the Mo-Sb colorimetric method, was $0.49 \pm 0.04$ g/kg in the P1, $0.89 \pm 0.02$ g/kg in the P2, and $1.07 \pm 0.05$ g/kg in the P3.

As in the Phase I experiment, we imposed two treatments in each environmental block: competition-free and competition (Fig. 1b). In the competition-free treatment, only one seedling of each species was grown in a separate pot. In the competition treatment, seven seedlings, each from a different species, were grown together in a pot. Healthy seedlings of similar initial heights were selected for each species (Table S7) and transplanted into the pots. The distance between seedlings within a pot was 3–5 cm. The pot size, spacing between pots, soil used, replacement of dead seedlings and insecticidal activity were all consistent with those used in the Phase I experiment. For each species in the competition-free and competition treatments in each abiotic environmental block, 45 replicate pots were established (360 pots per environmental block), resulting in a total of 3240 pots with 5670 seedlings. A total of 599 individuals with at least one missing trait value were excluded from the subsequent analyses in Phase II.

**Estimation of competitive rankings**
The competitive ranks among species were determined using the relative interaction intensity index (*RII*), which is a commonly used, unbiased estimation of competition intensity[73]. The smaller the *RII* is, the greater the level of competitive suppression experienced by the species. Specifically, for a group of species $i$ and $j$ in Phase I, the $RII_{ij}$ of species $i$ was calculated as $(\bar{B}_{ij,paired} - \bar{B}_{i,alone})/(\bar{B}_{ij,paired} + \bar{B}_{i,alone})$, where $\bar{B}_{ij,paired}$ and $\bar{B}_{i,alone}$ are the average individual dry biomasses of species $i$ in the competition and competition-free treatments, respectively. If $RII_{ij} \geq RII_{ji}$, species $i$ is less affected by species $j$ and is thus considered to have a greater competitive ability than species $j$. This means that in this group, species $i$ is competitively superior and species $j$ is inferior. Species $i$ and $j$ will retain the labels of superior and inferior species, respectively, for that species group in the competition-free treatment. We use the term 'competitiveness' to denote the competitive ability of species as determined by the above *RII*. Importantly, there were eight groups of species in Phase I, and a species could be either competitively superior or inferior in different groups, as competitive ability is not a constant attribute of a species and can be altered by different competitors and by resource availability.

For the Phase II experiment, the *RII* for species $i$ in environmental block $n$ was calculated as $(\bar{B}_{i,n,mixed} - \bar{B}_{i,n,alone})/(\bar{B}_{i,n,mixed} + \bar{B}_{i,n,alone})$, where $\bar{B}_{i,n,mixed}$ and $\bar{B}_{i,n,alone}$ are the average individual dry biomasses of species $i$ in the competition and competition-free treatments of the block, respectively. We calculated the *RII* of each species in each environmental block (Fig. S13). As *RII* values for the same species varied across different blocks, species competitiveness was determined

independently in each homogeneous environment. In heterogeneous environments, we determined species competitiveness by calculating the mean *RII* for each species in all nine blocks. To estimate the total biomass of individual seedlings (B), we harvested each seedling, including both aboveground and belowground parts, at the end of each experimental phase and dried them at 75 °C for 72 h to determine the total dry biomass (sum of aboveground and belowground dry weights).

Notably, the biomass-based *RII* was more appropriate than the methods based on relative abundance or frequency. This approach aligned with Begon's definition of competition, which he characterizes as an interaction among individuals competing for limited resources, leading to reduced growth, survival, and reproduction rates[11]. The *RII* estimates competition by comparing biomass, mortality or reproduction for a given species between the presence and absence of competition[73]. Experimental investigations often prioritize the seedling stage of trees, opting for biomass as a proxy for competition due to the challenges of obtaining mortality and reproduction data over short durations, such as a year[6,74,75]. On the other hand, the intensity of competition varies with the life history and abiotic environment of the plant, which causes it to be impossible to determine the competitive rank of two species in advance by pre-experiments or the distribution range of species in natural communities[76]. And the latter is affected by both competitive ability, soil seed bank, spatial and temporal heterogeneity, successional stage and resource situation[24,77,78]. This shows that species with high abundance in the community do not necessarily have high growth rates at the seedling stage. Therefore, in our seedling experiments, the competitive superior and inferior were defined after the experiment based on the final biomass, which represents the total competitive benefit of the plant.

To validate the competitive rank estimations derived from the *RII*, we also employed additional methods, such as the log response ratio (*lnRR*)[79] and the neighbor-effect intensity index with commutative symmetry (*NInt_C*)[80]. Given their consistent results with the *RII*, we exclusively report the findings from the *RII* for simplicity and clarity.

## Trait measurement

To ensure accurate estimation of the ITV[45], 45 replicates were performed for each species in the competition treatment in each environmental block. This process resulted in a total of 1080 and 5670 seedlings in the Phase I and Phase II experiments, respectively. After one year (two years for Phase II) of the experiment, we measured ten key functional traits for each seedling (Table S9) following a standardized protocol[81]. These traits included relative leaf chlorophyll content, leaf dry matter content, leaf mass per area, leaf toughness, leaf thickness, stem-specific density, stem moisture content, specific root length, specific root area and root tissue density. A detailed description of the trait measurements is given in Part 2 of the Supplementary Methods.

Specifically, relative leaf chlorophyll content (Chl) was measured on fully expanded leaves in situ using the Chl meter SPAD-502Plus (Konica-Minolta, Japan). The meter probe was placed at the adaxial leaf surface while avoiding pinching a mid-vein, measured three times, and averaged per leaf in each seedling. Chl of all seedlings was measured on the same day from 9:00 to 12:00 AM[82] (Marenco et al.[82]). Three leaves were randomly selected for leaf toughness (LTO) measurements using the "punch test" method[83] with a digital force gauge (precision 0.001 N, HADPI, Leqing, China). The induction needle (diameter is 1 mm) of the digital force gauge moved down at a constant speed of 10 mm·s⁻¹, piercing the middle position of each blade, and avoided primary and secondary veins. This was repeated three times at different positions, and then averaged the maximum force of three times as leaf toughness (GN) measurements were averaged. For measuring other leaf traits, twenty complete and healthy mature leaves were randomly collected for each seedling at 5 a.m. in the following days

and taken to the laboratory in a refrigerated incubator. The leaf thickness (LTh, mm) was first measured with a micrometer at five locations on the selected leaf. Then we immediately weighed all leaves to quantify leaf fresh weight (g) and measured leaf area (cm²) with a leaf area meter (LI-3000C, America). Finally, all leaves were dried to a constant weight in a well-ventilated oven at 75 °C as leaf dry biomass (g). Leaf dry matter content (LDMC) was the leaf dry biomass of an individual divided by its leaf fresh weight. Leaf matter area (LMA) was the ratio of leaf dry biomass to leaf area.

After measuring the leaf traits, all seedlings were harvested to measure the traits of stem and root traits. To estimate stem-specific density (SSD) and stem moisture content (SMC), one main stem was taken on 5 cm above the base of each seedling. Fresh volume was measured using the water displacement method within 24 h. Then the same section was dried for 72 h in a well-ventilated oven at 75 °C until a constant mass was achieved as stem dry weight. SSD was calculated as the ratio of stem dry weight and the fresh volume. SMC was defined as the difference between the fresh and dry weight of the stem divided by the dry weight of the same section. To measure root traits, we carefully cleaned the root surface from soil and attached organic particles with a 2 mm sieve under the shower head faucet until it was clean. Then, about 2 g of fine root samples less than 2 mm in diameter were used to scan fine root images (400 dpi) using a WinRhizo (Epson Expression 10000XL Scanner) flatbed scanner. Before scanning, the fine roots were scattered in trays without overlapping each other. Each fine root image was manually removed from background impurities (e.g., shadows and stains) using Adobe Photoshop (Adobe Systems), and was analyzed using the root analysis software WinRHIzO (Arabidopsis version 2012b, Regents Instruments Inc., Quebec Canada) to obtain total root length (cm), root area (cm²) and root volume (cm³) of the fine roots. Thereafter, each scanned root sample was placed in an envelope and dried to obtain its dry biomass (mg). Specific root length (SRL) and specific root area (SRA) were calculated as the ratios of total root length and root area to root dry biomass, respectively. Root tissue density (RTD) was the ratio of fine root dry weight to fine root volume.

Finally, the remaining aboveground and root (excluding the parts used to measure leaf, stem, and fine root traits) were dried in an oven for 72 h at 75 °C, and their corresponding dry biomass for each seedling was obtained. Total shoot biomass was the sum of the dry weights of the leaves, stem, and remaining aboveground parts, and total root biomass was the sum of the fine roots and remaining root.

## Estimation of intraspecific trait variability

The ITV of each species was quantified using two different approaches: a multidimensional trait approach and an individual trait approach. The first emphasizes the importance of trait covariance in determining species fitness[41–43,84], while the latter highlights the unequal roles traits play in fitness[6,7,44]. Both approaches have unique strengths and can offer different insights into the relationship between ITV and species competitiveness. Specifically, the ITV of multidimensional traits was quantified using the size of the multidimensional trait space, also known as hypervolume[85,86]. To estimate the hypervolume for each species, we used the Silverman bandwidth estimator and applied a 5% quantile threshold to create a multidimensional Gaussian kernel density. To ensure robust estimation of hypervolumes and fair comparisons between hypervolumes, we employed two methods. First, we used seven functional traits (Table S9) with correlation coefficients less than 0.8 (Fig. S14) to estimate hypervolume in both the Phase I and II experiments. Additionally, we estimated a reduced-dimensional hypervolume based on the scores of the first three principal component axes according to the aforementioned traits (Table S10 and Fig. S15)[87]. The number of principal components required was determined using Horn's parallel analysis[88].

To robustly compare hypervolumes between species across treatments, we first calculated 999 hypervolumes for each species

based on random samples of 20 conspecifics for Phase I and 15 conspecifics from either a single homogeneous environment (Fig. 2b, e) or nine different environments with varying conditions (Fig. 2c, f) for Phase II. This process was performed separately under both competition-free and competition treatments using the same method as that used by Bittebiere et al.[47]. We then took the mean and standard error of the 999 hypervolume values to obtain the mean and variability in the ITV for each species under each scenario (e.g., competition-free homogeneous, competition-free heterogeneous, etc.) (Fig. 2). All trait values were centered and scaled for all individuals across all pots in both Phase I and Phase II, as recommended by Blonder[86]. The sizes of all hypervolumes in this study are reported in units of standard deviation raised to the power of the number of trait dimensions used, i.e., $SD^7$ and $SD^3$ for the seven-dimensional and the reduced three-dimensional PCA hypervolumes, respectively.

In addition, an individual trait approach was also used to quantify the effect of competition on intraspecific variability for each trait. We first calculated the intraspecific variability for each trait in each species using the improved coefficient of variation (*Bao's CV*)[45]. We then calculated both the absolute and relative changes in ITV from competition-free to competition treatments across the three competitive scenarios (pairwise competition in a homogeneous environment and multispecies competition in homogeneous and heterogeneous environments). A positive change in ITV indicates that the relative variability of the target trait increased under competition; a negative change indicates a decrease in the relative variability of the target trait. In Phase II, we deliberately selected seven (with correlation coefficients less than 0.8) out of ten traits to calculate the ITV, while in Phase I, we used all ten traits. This selection aimed to make the multidimensional trait space comparable between phases. To test whether trait selection introduced bias in Phase II, similar analyses were performed using all the functional traits.

**Comparison of intraspecific trait variability between competing species**

For Phase I, we used the Paired Wilcoxon rank-sum test to examine whether competitively inferior species had higher ITVs than superior species did, separately, in the competition-free and competition treatments. In Phase II, we tested the relationship between ITV and competitiveness (*RII*) and whether this relationship differed between competition treatments, separately, in spatially heterogeneous environments and in environmental block 1, which had the same homogeneous environment as in Phase I, allowing comparison of results between the two phases. This was done by fitting mixed-effects models with competition treatments (competition-free and competition), competitiveness (*RII*), and their interaction as fixed effects and species as a random effect. To confirm the robustness of the results from block 1, we expanded our models to include all nine homogeneous environmental blocks shown in Fig. 1c, with species and environmental blocks as two crossover random factors. We quantified the ITV and *RII* in spatially heterogeneous environments using aggregated populations from the nine environmental blocks (Fig. S12). To evaluate the potential bias of life forms, we performed similar analyses after excluding the two deciduous species.

To quantify the effect of competition on ITV, also known as competition-induced intraspecific variability, we calculated the absolute and relative changes in multidimensional ITV for each species from the competition-free to competition treatments in both Phases I and II. These changes were compared to the *RIIs* of competing species using a simple linear regression model. Similar analyses were used to test the relationship between changes in individual trait variability and species competitiveness. The statistical analyses in this study were performed using R software (v.4.0.5; R Core Team, 2019), and the R package hypervolume (v.3.1.1) was

used to calculate the multidimensional hypervolumes from the trait data[89].

**Reporting summary**

Further information on research design is available in the Nature Portfolio Reporting Summary linked to this article.

## Data availability

The data that support the findings of this study are available on the Figshare digital repository (https://doi.org/10.6084/m9.figshare.24174558.v5).

## Code availability

The code supporting all results are available on the Zenodo (https://doi.org/10.5281/zenodo.10809719) and GitHub (https://github.com/Jingyangecnu/Inverse-relationship-between-species-competitiveness-and-intraspecific-trait-variability.git).

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

## Acknowledgements

We are grateful to Prof. Fangliang He for his constructive comments on the manuscript. We thank Qiuwu Yu, Siyu Wu, Congling Zhang, Ruijiao Jiang, Jiahui Lu, Yue Chen and many other people who helped implement the experiment and collect data. G.C.S. was supported by the National Natural Science Foundation of China (NSFC) (32271596), Natural Science Foundation of Shanghai (NSFS) (23ZR1419200), "Pioneer" and "Leading Goose" R&D Program of Zhejiang (2023C03137) and other grants (2023JBGS01, 2021ZDZX01, 2021ZDLY03). J.Y. was supported by the fellowship of China Postdoctoral Science Foundation (2022M721160) and National Natural Science Foundation of China (32301335). C.P.C. was supported by the Estonian Research Council (PSG293 and PRG2142) and the European Regional Development Fund via the Mobilitas Pluss program (MOBERC40 and MOBERC100).

## Author contributions

J.Y. and G.C.S. designed the pot experiment. J.Y. and X.Y.W. collected the data. J.Y. and G.C.S. conceived the research questions. J.Y. conducted the analyses and wrote the first manuscript draft with input from G.C.S., C.P.C. and X.H.W. All authors contributed to revising the manuscript.

## Competing interests

The authors declare no competing interests.
