## [Peer Review File · Nature Communications]

Inverse relationship between species competitiveness and intraspecific trait variability may enable species coexistence in experimental seedling communitiesReviewer #1 (Remarks to the Author):

This study uses a series of experiments to assess whether the “intraspecific mean-variance trade off” that has been explored with theoretical studies emerges in empirical work. The size of the experiment is impressive and the theoretical framework for this investigation is solid. The results are noteworthy because the authors do in fact demonstrate this trade-off with their experimental work.

However, after reviewing the manuscript several times, I am still confused about why and how specific approaches were taken and wary of the conclusions made in relation to coexistence. Considerable restructuring and clearer justification for specific approaches and experimental decisions is needed to make the manuscript understandable and appealing to a broad ecological audience. My main concerns apply throughout all sections of the manuscript and include: (1) Confusion about what exactly the “interspecific mean-variance trade-off” is. This trade-off needs to be more clearly defined. In providing this information in the introduction, it would be helpful to provide information about how this trade-off can be explored and tested empirically prior to introducing specific hypotheses associated with the study. (2) The experimental design is confusing to me. Specifically, how and why are the heterogeneous environments included? Why would someone expect the trade-off to emerge in heterogeneous environments but not in a homogeneous one? Are the specific treatments (i.e., light, soil moisture, phosphorous) in the heterogeneous environment comparable and is grouping these results together reasonable? More information for this experimental decision and associated analyses is warranted. (3) I am skeptical that the experimental design and results of this work actually inform our understanding of coexistence even though implications for coexistence are implied throughout the manuscript. Justification for why competition studies with the specific performance metrics used (i.e., biomass) are reasonable to infer coexistence is needed. Finally, (4) I find parts of the manuscript tangential to the core ideas of this work. For example, IV resulting from adaptations vs. plasticity are discussed in the introduction but not mentioned again in the manuscript. I recommend the authors narrow in on the core questions, core results, and eliminate text related to ideas that cannot be tested and do not add to the overarching message of the work. Specific line comments are below.

Lines 37 – 38: What is the competitive response of a trait? Because this term is frequently used in competition literature in regard to performance outcomes of competition, its use here is confusing. To me, if a trait is responding to competition, this may be assessed as IV resulting from plastic responses to a stressor (i.e., competition in this case), but it is not a “competitive response” in performance outcomes.

Lines 53 – 56: Given the IV is a population or species-level metric, I am unclear about how it affects competitiveness of *individual* members of a species? Here and throughout the manuscript, I think it is important to note that outcomes of pairwise or group competition trials may not directly translate coexistence outcomes. Long-term population growth rates would be needed to make these claims.

Lines 63 – 66: This is a bit unclear. Is the IV for a specific trait being measured, is it multi-dimensional, or does it apply to a performance outcome as suggested here (i.e., seed production)? This ought to be defined early in the manuscript.

Line 68: As far as I can see, Bolnick et al. 2011 does not refer to differences in competing species IV as the “interspecific mean-variance trade-off”. Because of this omission, I am having a tough time conceptualizing this trade-off. Please explicitly define the trade-off and provide citations for the use of this term that readers can refer to.

Lines 70 – 71: The second part of this sentence seems tangential. If stressful conditions are to be explicitly tested this ought to be discussed to a greater extent in the introduction and related to the expected patterns.

Lines 75 – 76: Is the effect or impact of this trade-off for coexistence also absent? If yes, it is worth

stating that here.

Lines 77 - 110: I appreciate the discussion of adaptations vs. plastic responses resulting in IV. But, within the present study the authors do not determine whether IV results from adaptations (i.e., genetic differences among individuals within a population), maternal effects (i.e., differences in traits resulting from the maternal environment), or plasticity. I also do not think it is possible within the context of the study. If this text is to be included, it ought to be more directly related to the current study and the implication of these difference origins of IV relation to competition and coexistence ought to be discussed.

Lines 129 – 132: This is actually two questions and I think they ought to be listed individually. Also, please clarify when the environment is homogenous vs. heterogenous for each of these questions.

Lines 132 (and above): Please discuss how IV resulting from heterogenous environments vs. competition are related.

Line 137: Trade-off is misspelled.

Lines 148 – 155: I am concerned about the potential for maternal effects with this design. Plants could express traits related to the maternal environments in which they are grown, so assessing whether IV results from adaptations vs. plasticity is difficult if seeds from a common environment are not used in competition studies. If the origin of trait differences is not relevant or does not affect results, this ought to be mentioned.

Line 171: Given that this section also describes the heterogenous environment component of the study, I think this ought to be reflected in the section heading.

Lines 194 – 195: This is confusing given the various abiotic treatments described above. Please clarify.

Lines 239 – 241: Were both aboveground and belowground biomass collected? Could results differ if belowground biomass was also assessed. Please discuss limitations here and throughout.

Lines 261 – 263: I am not convinced that using hypervolumes is the most appropriate approach to assessing the effects of IV on competition since some traits, but not others, can be responsible for competitive outcomes among species. See Kraft et al. 2015 PNAS <https://doi.org/10.1073/pnas.141365011> and Funk & Wolf 2016 Ecology <https://esajournals.onlinelibrary.wiley.com/doi/full/10.1002/ecy.1484> for examples. If the authors choose to retain these analyses, justification for using hypervolumes or other aggregate measures of multi-dimensional trait variability in relation to competition ought to be mentioned in the introduction.

Line 309: How and why are results combined from from the heterogenous environments? Do species vary in traits that allow them to be more or less successful in these different treatments? If yes, could this bias results?

Lines 466 – 478: I'm wary of the discussion related to the directionality of changes in traits since the results presented do not mention directionality of trait changes, just changes in IV. The discussion ought to more directly relate to the results presented.

Lines 487 – 489: Given that competition intensity does not necessarily translate to long-term coexistence, I think this conclusion is a bit of a stretch here and throughout the manuscript.

Reviewer #2 (Remarks to the Author):

Overall comments

This manuscript seeks to clarify how intraspecific trait variation can mediate competitive species interactions (and coexistence), and how IV might vary in response to the environment. This is an important area of research that is largely overlooked, and the authors use an extensive dataset of seedlings to address their questions. I struggled to follow the paper conceptually, primarily due to the vagueness of the text and the lack of clear examples. The main hypothesis, that inferior species should have more IV than superior species, is not an intuitive one, and the authors should explain the mechanism in more detail as it would seem that the amount of variation should depend on the trait in question.

For example, when there is strong competition for light, I would expect strong directional selection for taller stature in the superior species (and possibly less IV), whereas I would expect no influence on IV in height among the inferior species. The inferior species may instead grow laterally, or make thicker leaves with more photosynthetic tissue, or alter the leaf angle, such that there is no impact on height. I was looking for more explanation and examples of this sort within the text. But it seems as though it was written in a very broad way so as to describe trait variation in general, which I don't think worked well.

Apart from the conceptual issues, I had many questions about the methods and the experimental design, which was also lacking some detail. I thought the figures were very well made as were the tables, but again, I really struggled with the conceptual side of this paper, and I wasn't fully convinced that IV should be higher in the inferior species, at least not without more examples from their own study and from previous studies. The one section of the paper where they do provide examples using below- and aboveground traits is also incorrect in that they incorrectly assert that higher root tissue density means a more acquisitive strategy, when it is actually the opposite. And there's one short sentence that notes that the patterns of variation in IV will depend on the trait in question.

Line by line edits

Line 35 How is an inferior or superior species defined? I did go to the appendix eventually but it seems it should be better defined early on.

Line 37: how is a competitive response evaluated? In terms of a reduction in plant size? For example?

Line 39: is it a greater increase in IV or the amount of IV itself? If it's the former, then that's slightly different from the theory and perhaps needs clarification.

Line 43: incomplete sentence

Line 63: this sentence is a bit unclear. When two species have the same IV, the superior species is defined as that which benefits more from having a wider trait space (or possibly ecological niche?). I don't understand how the increased per capita seed production comes into play. Because a wider trait space would mean that some individuals have lower trait values and some have higher trait values, in other words often a greater spread of values. I suppose the key point here is that it's not just about the mean trait values but also the spread around those values that determines the strength and direction of the interaction. I think this paragraph could be clearer.

Line 78: between competing species or between species in general? It's an important distinction as far as theory.

Line 79: suppression in terms of plant size? Survival? Reproduction? I just want to know how this is defined.

Line 90: I didn't follow this sentence; can you explain further?

Line 95: unclear why 3 years was used as an example. What trait was measured in these example studies that are referenced? And some species don't even live 3 years so again, the example seems arbitrarily chosen. Some traits change within seconds to minutes.

Line 98: unclear what is meant by different changes in IV between competing species. Do you mean the sensitivity of IV could vary across species? That's my interpretation based on the following sentence.

Line 101: I don't see how one results from the other. I don't follow the logic here. Just because a

species is an inferior competitor doesn't mean it's going to experience more (or less) trait variation. More explanation is needed to substantiate this argument.

Line 103: vague

Line 107: can you provide an example? For example, plant height is a trait that would vary in response to increasing abundance/density of neighbouring plants (whether the same species or different species). But height wouldn't necessarily become more variable, if there is strong directional selection to become taller in response to more competition.

Line 144: please add the country

Line 163: approximately how large were the seedlings at the time of transplant? I think the seedling size at the beginning should have a strong impact on future competitive ability. Were these species reared in high light, high moisture, high nutrients? There's almost no information about the environmental conditions the plants experienced.

Line 168: how far apart were the seedlings? If the seedlings were quite small, they would have weak competition but if one was larger than the other, it would be asymmetric. Note: this information is presented on line 205 and should definitely be moved up sooner.

Line 181: insufficient detail regarding the environmental conditions. What was the mean and maximum for each environmental property? Providing the minimum light doesn't tell us much about potential light stress. I see that an appendix is referenced but I don't think we should need to see the reference for this very basic information about growth conditions. Even just a few sentences would be useful.

Line 196: can we also know the range of variation in the environmental variables over the course of a day or months? The term homogeneous is somewhat subjective as it depends on the time frame.

Line 222: it would have been helpful to know earlier that competitive effects were based on biomass

Line 225 and line 236: the authors say this is to avoid confusion yet it is very confusing. The definition of RII is defined in terms of biomass when a species is within a pair, yet this line says that you use the term competitiveness even when the species is not in a pair. So does this mean that when I see the terms superior or inferior, that this species may not necessarily be in a pair? That it could be on its own? Because then later you note how the characterization of a species as being superior or inferior depends on the identity of the other species within the pair.

Line 230: and by resource availability

Line 279: is it possible to also calculate the amount of variance around the mean hypervolume? Based on Figure 2 it looks like you have the standard error

Line 407: I disagree that this is the case for all traits. That's my main argument. That some traits will tend to be quite variable within species even in the absence of competition or heterogeneity. For example, leaf size is hugely variable in some species due to leaf ontogeny.

Line 418: optimal being defined in terms of max biomass?

Line 418: a concave down trait performance curve but for which trait(s)? I don't see how you can generalize so broadly.

Line 420: why would an individual produce a suboptimal trait value in response to a cue? Are you suggesting that an inferior species in low moisture, might have some individuals that make deeper roots and others that make shallower roots, even if the more shallow-rooted individuals perish? Is this part of the reason why these species are inferior in the first place? Are you suggesting this is maladaptive? It seems that there should be more discussion about alternative resource use strategies and performance landscapes (sensu Laughlin).

A concave down trait performance curve versus a concave down doesn't tell us anything about the extent of variation though. They could have different shapes but the exact same amount of variation, so the sentence that follows from this (Line 424) doesn't exactly follow the logic (in my opinion).

Line 469: negative or positive change? Add + or -

Line 475: it does not make sense to me why a species would be conservative above and acquisitive below. I think the use of those terms is not helpful in this context. Higher LDMC could be associated with longer leaf lifespan, and higher stem density usually helps to ameliorate water stress, so the higher root tissue density should be a drought adaptive response, not evidence of a more resource acquisitive strategy. Root tissue density also typically increases with a reduction in soil nutrients, so again this isn't evidence of being more acquisitive on bottom and conservative on top.

Reviewer #3 (Remarks to the Author):

This study explores differences in intraspecific variability between competitively inferior vs superior tree species in order to test for an interspecific mean-variance trade-off. The authors use an experimental approach that clearly involved a massive amount of work and is impressive in its scope. The results provide novel experiment evidence of the mean-variance trade-off, which is relevant for species coexistence. Overall, the study makes a nice contribution to our understanding of the role of intraspecific variation in structuring diverse communities.

My main concern about the study is that the authors provide no information about where or how the seeds used in the experiment were collected (e.g., over what spatial extent, how many source trees were used, etc). This information is critical for any study of intraspecific variability (IV). If competitively inferior species were rarer in the community and/or produced fewer seeds than competitively superior species, then there might be the need to collect seeds of inferior species from more source trees and over a larger spatial extent, which could bias estimates of IV.

On a related note, the authors appear to be using the term IV to mainly refer to phenotypic plasticity. However, IV can arise due to genetic variation and phenotypic plasticity (and there can also be genetic variation in phenotypic plasticity). For the authors' first question about IV in a competition-free, homogenous environment, then the IV would be the result of genetic variation (since you would not expect to see plasticity under uniform conditions). For questions 2 and 3, about IV under competition and in heterogenous abiotic environments, then they are talking about phenotypic plasticity. The authors need to be clearer about the underlying drivers of IV (eg genetic variation and phenotypic plasticity). This is related to my comment above about seed collection, since that will influence how much genetic variation exists in the seed pool used in the experiment as well as how plastic the collected individuals are expected to be (eg if some species were collected from more heterogenous environments than other species).

My other concern is about the influence of growth form on the results. Specifically, the two deciduous species included in the experiment were the most competitively superior in both Phase 1 and Phase 2 experiments. Deciduous species employ a different ecological strategy than evergreen species, which typically means that they differ in other traits as well. Do the observed relationships remain qualitatively similar if removing those two species from the analyses? (I don't mean whether they are still statistically significant because they probably won't be just due to the smaller sample size, but rather whether the trend is the same.)

Some additional comments:

- Line 65: Why just "per capita seed production"? Wouldn't any increase in fitness have this effect?
- Line 69: add "the" before "same"
- Line 71: add "the" before "Advantage"
- Line 104: It is not clear here why this would lead to large IV in only inferior species (and not also in superior species)
- Line 106-107: I don't follow the logic here – why would it deviate from its optimal trait value. If IV is increased to enhance a species advantage, presumably it is achieving the optimal trait value for a given environment.
- Line 126: do these 10 tree species coexist in nature? Are they all native?
- Line 144: indicate the country where the site occurs (China)
- Line 149: What was the rationale for selecting both evergreen and deciduous (and 11 of one type but only 6 of the other type)? Was there some expectation about how they would differ?
- Line 151: As mentioned above, there needs to be a lot more detail about seed collection (how many mother trees, over what geographic area and types of environments, were all species collected at the same sites, etc).

- Line 158: What kind of homogenous environment? What resources were the species likely competing for in that environment (e.g., where soil nutrients limited, was there limited light or water, etc)?
- Line 180: It is not clear here or anywhere in the main text or supplement how the 9 different environments were selected. It was not all combinations of the 3 variables, so why were some combinations excluded?
- Line 198: "populations of two competing species" is confusing here because competition was not pairwise in Phase II
- Line 277: Was this done separately for competition-free and competition treatments?
- Line 279: The hypervolume values were averaged separately for each species under each scenario (e.g. competition-free homogenous, competition-free heterogenous, etc)?
- Line 280: State how many individuals had missing trait data
- Line 288: was the change in IV calculated in absolute or relative (ie % difference) terms? Looking at the graphs, it appears to be absolute differences. Do the results change if using % difference instead?
- Line 292-294: Why was this only done for Phase II?
- Line 303-305: Why wasn't species included as a random effect? It sounds like you have multiple values for each species in the model (eg one for each treatment), but are not accounting for the fact that values from the same species are not independent of each other.
- Line 308: Was this done using data from treatments with or without competition?
- Line 312: Using Phase I or Phase II data (or both)?
- Line 310-314: Why didn't you model IV as a function of competition, RII, and RII x competition interaction (with species as a random effect)? That would be a more straightforward test of 1) whether competition affects IV and 2) whether the relationship between RII and IV changes under competition. If you model it that way, are the results consistent?
- Line 332: change "survival" to "surviving"
- Line 337: specify that "significant suppression" is referring to reduced biomass.
- Line 369-370: There is a typo here in the numbers – they do not match up with what is on the graphs (and also how can you have an R-square value of -8.7).
- Line 380: When it says "individual traits", it makes it sound like you ran separate models for each trait. But instead, it seems like this is just a single model for all traits. This needs to be better explained both here and in the methods.
- Line 382: Is this in the homogenous environment only? (Both here and throughout the Results, make sure to specify whether you are referring to homogenous vs heterogenous environment when referring to competition treatment results. It is often not clear.)
- Line 387: Specify "greatest" compared to what (ie other traits measured).
- Line 398: change "seedlings stage" to "seedling stage"
- Line 407-409: How exactly are determining that the traits are optimal? It seems like you are assuming that they are because of the low IV, but low IV does not necessarily mean optimal trait values! Do you also see lower interspecific variation? The fact that you find low IV in a homogenous environment just means that there is little genetic variation in the trait in your seed pool, and that increased IV under heterogenous environments and competition is resulting from phenotypic plasticity.
- Line 411-413: This makes no sense to me. If IV is selected for by the environment, then presumably individuals would be accurately perceiving the environmental stresses and adjusting trait values accordingly to increase fitness.
- Line 414 and Line 420: specify what trade-off you are referring to
- Line 417-422: This section is confusing and hard to follow for anyone who is not already very familiar with the papers being reference. Please rephrase to explain this more clearly.
- Line 431: remove the word "only". Previous studies have also found that IV is related to genetic variation, species life history strategies, dispersal modes, etc.
- Line 460-461: rephrase as "which could allow invasive species to survive in the face of...". As written, it makes it sound like invasive species are actively choosing this strategy, rather than high plasticity being one of the reasons they become invasive.
- Line 466: was this higher IV in root traits compared to leaf and stem traits found in all environments? What about if you just look at environments that differ in light only? I would expect above ground

traits to respond more to differences in light, and root traits to respond more to differences in soil moisture and nutrients. Given that 2 of the 3 abiotic variables being manipulated in the experiment were below ground variables, it is not surprising that root traits would appear to have higher IV. You need to consider the specific abiotic variable when comparing the different traits.

-Figure 1: again, it is not clear how the 9 environments were chosen since it is not all combinations of the 3 variables.

-Figure 2 and Figure 3: Include exact P values for significant relationships, not just $P < 0.05$

-Figure 2, line 692: Do you mean "orange circles, b-d" here? I think it is a typo because only d has orange circles.

-Figure 3: something is off with the figure legends. For example panel b legend has orange and blue lines for homogenous environment, but there are no orange and blue lines in the graph. Also in panel A, it is not clear what the difference is between blue and purple.

Figure 4: Is this from a homogenous environment only? Specify in the legend.

Response letter to the reviewers' comments

Reviewer #1 (Remarks to the Author):

This study uses a series of experiments to assess whether the “intraspecific mean-variance trade off” that has been explored with theoretical studies emerges in empirical work. The size of the experiment is impressive and the theoretical framework for this investigation is solid. The results are noteworthy because the authors do in fact demonstrate this trade-off with their experimental work.

Response: Thank you for your positive feedback on our study.

However, after reviewing the manuscript several times, I am still confused about why and how specific approaches were taken and wary of the conclusions made in relation to coexistence. Considerable restructuring and clearer justification for specific approaches and experimental decisions is needed to make the manuscript understandable and appealing to a broad ecological audience. My main concerns apply throughout all sections of the manuscript and include:

1. Confusion about what exactly the “interspecific mean-variance trade-off” is. This trade-off needs to be more clearly defined. In providing this information in the introduction, it would be helpful to provide information about how this trade-off can be explored and tested empirically prior to introducing specific hypotheses associated with the study.

Response: Our apologies for the lack of clarity in explaining the idea of the trade-off. We have much revised this part (lines 52-61) to better convey the intended meaning. Specifically, the ‘mean’ in the trade-off refers to the mean competitive ability among conspecifics of a species, while ‘variance’ refers to trait variability (e.g., variance of functional traits) of the species. The interspecific mean-variance trade-off indicates that species with the lower mean competitive ability (i.e., inferior species) tend to exhibit greater intraspecific trait variation. This trade-off was first introduced by Begon & Wall (Begon & Wall 1987) and formally defined by Lichstein *et al.* (2007). It has subsequently gained theoretical support from several theoretical studies (Crawford *et al.* 2019; Des Roches *et al.* 2018; Feniova *et al.* 2013; Hart *et al.* 2016; Milles *et al.* 2020; Uchmański 2021; Uriarte & Menge 2018).

The methodology for testing this trade-off is now summarized in lines 107-121. Specifically, we estimated species’ mean competitive ability for a given competition system by growing seedlings of different species together in pots and comparing their biomass between competition and competition-free treatments (see lines 463-470 and 476-479 for details). These approaches enable us to identify species as superior or inferior based on a relative competition index (*RII*) calculated for seedling biomass. Meanwhile, ITV of each species was assessed via the size of multidimensional trait space and individual trait variability, respectively. Lines 505-511 and 535-537 illustrate these methodological details.

2. The experimental design is confusing to me. Specifically, how and why are the heterogenous environments included? Why would someone expect the trade-off to

45 emerge in heterogenous environments but not in a homogenous one? Are the specific
treatments (i.e, light, soil moisture, phosphorous) in the heterogenous environment
comparable and is grouping these results together reasonable? More information for
this experimental decision and associated analyses is warranted.

**Response:** Inclusion of heterogeneous environments stems from recognizing that
individuals of the same tree species often grow across diverse abiotic conditions. Such
heterogeneity can alter resource availability and the niche space of the species,
potentially changing the values of many functional traits and the magnitude of ITV
(Banitz 2019; Uriarte & Menge 2018). Heterogeneous environments could also shift
competitive hierarchies among species compared to homogeneous environments
(Wang & Callaway 2021). Elucidating these intricate effects of heterogeneity on the
trade-off is a big challenge, with unclear theoretical expectations. Thus, one aim of our
study was just to explore whether relative ITV magnitude among competing species
differs between homogeneous and heterogeneous environments. We did not expect the
trade-off to exist only in heterogeneous environments. We have now clarified the
rationale for incorporating heterogeneous environments in lines 98-104 in the
introduction and discussed potential mechanisms underlying the observed trade-off
difference between homogeneous and heterogeneous environments (lines 260-273).

Heterogeneous environments were incorporated by creating pots with distinct
abiotic conditions (see Fig. S10 and Table S6 for details). For instance, pots in block 8
had low soil moisture and phosphorus under medium light, while pots in block 1 had
high soil moisture, low soil phosphorus and high light. These distinct pot conditions
mimicked conspecific individuals experiencing diverse heterogeneous environments
across our study area. We have reworded the heterogeneity manipulation description
(lines 414-417, 419-445) to more clearly articulate the methodology.

We agree that the interspecific mean-variance trade-off may not be directly
comparable across different abiotic environments. Our study did not intend to make
such cross-environment comparisons. Instead, following previous studies (e.g.,
Allesina & Levine 2011), we considered conspecifics across abiotic environments as
one population and calculated their species-level ITV and competitiveness (*RII*). We
now add a new Fig. S10 that builds on Fig. 1 to further illustrate details about the abiotic
environmental conditions, competitive treatments, and species definitions in both
homogeneous and heterogeneous environments in phase II experiment. We then
explored the relationship between species-level *RII* and ITV under competition and
competition-free treatments, respectively. This allowed us to test the interspecific
mean-variance trade-off within spatial heterogeneous environments. A more detailed
method description has been added in lines 414-417.

3. I am skeptical that the experimental design and results of this work actually inform
our understanding of coexistence even though implications for coexistence are implied
throughout the manuscript. Justification for why competition studies with the specific
performance metrics used (i.e., biomass) are reasonable to infer coexistence is needed.

**Response:** This is an important yet challenging question. As far as we know there is no
proven metrics that can be claimed to genuinely measure competition. Biomass (or

productivity), species abundance, and demographic (growth and mortality) rates are
among those that are often used to measure competition in the literature. Our study
follows this common practice by using biomass to represent performance of the plants,
but admittedly we cannot guarantee biomass is indeed the best measure. Another
uncertainty is that it is difficult to know how long the experiment has to take for
assessing competition of tree species. Because of these issues, we focused our study on
ascertaining whether there are trait ITV trade-offs between inferior and superior species.
Such evidence serves as a necessary (but may not be sufficient) prerequisite for
coexistence. Thus, the existence of the trade-off would just suggest the possibility of
coexistence. Following this comment, we have now discussed this issue (lines 315-319),
explaining the reason why seedling biomass was used to measure competitiveness (see
Part 2 of Supplementary Methods). We also lowered down the tone for arguing
coexistence throughout the text.

4. Finally, I find parts of the manuscript tangential to the core ideas of this work. For
example, IV resulting from adaptations vs. plasticity are discussed in the introduction
but not mentioned again in the manuscript. I recommend the authors narrow in on the
core questions, core results, and eliminate text related to ideas that cannot be tested and
do not add to the overarching message of the work.

**Response:** Thank you for this valuable suggestion. We have narrowed the scope by
removing the discussion about adaptation and evolution. We have revised the study to
focus on our central questions and findings (lines 65-73).

**Specific line comments are below.**

5. Lines 37 – 38: What is the competitive response of a trait? Because this term is
frequently used in competition literature in regard to performance outcomes of
competition, its use here is confusing. To me, if a trait is responding to competition,
this may be assessed as IV resulting from plastic responses to a stressor (i.e.,
competition in this case), but it is not a “competitive response” in performance
outcomes.

**Response:** Agreed. The term "competitive response" has been replaced by "plastic
response" (line 27).

6. Lines 53 – 56: Given the IV is a population or species-level metric, I am unclear
about how it affects competitiveness of *individual* members of a species?

**Response:** It was really meant “competitiveness of species”. The sentence was so
revised on line 43.

7. Here and throughout the manuscript, I think it is important to note that outcomes of
pairwise or group competition trials may not directly translate coexistence outcomes.
Long-term population growth rates would be needed to make these claims.

**Response:** We fully agree that long-term population growth is needed to predict
coexistence outcome. We have added relevant evidence of ITV on long-term population

growth on line 44 and discussed this limitation of our experiment on lines 315-319.

8. Lines 63 – 66: This is a bit unclear. Is the IV for a specific trait being measured, is it
multi-dimensional, or does it apply to a performance outcome as suggested here (i.e.,
seed production)? This ought to be defined early in the manuscript.

**Response:** Sorry for the confusion. In the theoretical work of Hart (2016), ITV was
defined as variation of a single model parameter such as competitive sensitivity (line
53). In our study, the definition of ITV was based on empirically measured trait values
from both one-dimensional and multidimensional trait space (lines 113-115).

9. Line 68: As far as I can see, Bolnick et al. 2011 does not refer to differences in
competing species IV as the “interspecific mean-variance trade-off”. Because of this
omission, I am having a tough time conceptualizing this trade-off. Please explicitly
define the trade-off and provide citations for the use of this term that readers can refer
to.

**Response:** Sorry for the lack of clarity here. We have defined the interspecific mean-
variance trade-off in the revision in lines 52-61 and provided citations on line 59.

10. Lines 70 – 71: The second part of this sentence seems tangential. If stressful
conditions are to be explicitly tested this ought to be discussed to a greater extent in the
introduction and related to the expected patterns.

**Response:** For simplicity and clarity, we have decided to narrow the focus to the
relationship between ITV and species competition. Thus, discussions related to stressful
abiotic conditions have been removed.

11. Lines 75 – 76: Is the effect or impact of this trade-off for coexistence also absent?
If yes, it is worth stating that here.

**Response:** Yes, as far as we know, the direct empirical test of this trade-off for
coexistence is absent. However, answering this question is beyond the capacity of the
current experiment, because it would need a more complex design (e.g., response
surface design) and longer experiment for tree species. Therefore, we only focus on the
existence of the trade-off itself but have highlighted this unresolved issue as a direction
for future research in the Discussion (lines 317-319).

12. Lines 77 - 110: I appreciate the discussion of adaptations vs. plastic responses
resulting in IV. But, within the present study the authors do not determine whether IV
results from adaptations (i.e., genetic differences among individuals within a
population), maternal effects (i.e., differences in traits resulting from the maternal
environment), or plasticity. I also do not think it is possible within the context of the
study. If this text is to be included, it ought to be more directly related to the current
study and the implication of these difference origins of IV relation to competition and
coexistence ought to be discussed.

**Response:** We agree that our experiment cannot determine the sources of ITV.
Following this comment, we have removed discussion about the causes of ITV from
the Introduction section.

13. Lines 129 – 132: This is actually two questions and I think they ought to be listed
individually. Also, please clarify when the environment is homogenous vs.
heterogenous for each of these questions.

**Response:** Revised accordingly on lines 119-125.

14. Lines 132 (and above): Please discuss how IV resulting from heterogenous
environments vs. competition are related.

**Response:** Heterogeneous environments and competition are closely related in
determining ITV magnitude. Specifically, competition may induce plastic trait changes
that increase ITV, while the abiotic environment may constrain these changes by
filtering out unfit trait values induced by competition. Heterogeneous environments are
expected to have less constraint on traits than homogeneous environments. In addition,
heterogeneous environments may alter the intensity of competition between species,
which in turn has the potential to indirectly affect competition-induced trait changes
and ITV. We have added this information on lines 98-104.

15. Line 137: Trade-off is misspelled.

**Response:** Corrected.

16. Lines 148 – 155: I am concerned about the potential for maternal effects with this
design. Plants could express traits related to the maternal environments in which they
are grown, so assessing whether IV results from adaptations vs. plasticity is difficult if
seeds from a common environment are not used in competition studies. If the origin of
trait differences is not relevant or does not affect results, this ought to be mentioned.

**Response:** Maternal effects are not fully controlled in our experiment, limiting our
ability to precisely distinguish the effects of adaptations and plasticity on ITV.
Therefore, we have removed the adaptation and plasticity discussions from the
Introduction and explicitly recount the limitations in Discussion (lines 220-226). All
that said, we believe maternal effects was not strong in our experiment, because all
seeds for each species were collected from a single healthy mother tree and all mother
trees were located within a 1 km² area with similar abiotic environments. We have
added more details about seed collection on lines 349-353.

17. Line 171: Given that this section also describes the heterogenous environment
component of the study, I think this ought to be reflected in the section heading.

**Response:** Done.

18. Lines 194 – 195: This is confusing given the various abiotic treatments described
above. Please clarify.

**Response:** To improve the clarity, we have revised this sentence (lines 413-417).

19. Lines 239 – 241: Were both aboveground and belowground biomass collected?
Could results differ if belowground biomass was also assessed. Please discuss
limitations here and throughout.

**Response:** Both aboveground and belowground portions were harvested for each
seedling. We have revised the text (lines 483-487) to clarify that total seedling biomass
(aboveground + belowground) was used for calculating relative competitive index. In
addition, we stated the above information earlier on lines 113-115.

20. Lines 261 – 263: I am not convinced that using hypervolumes is the most
appropriate approach to assessing the effects of IV on competition since some traits,
but not others, can be responsible for competitive outcomes among species. See Kraft
et al. 2015 PNAS <https://doi.org/10.1073/pnas.141365011> and Funk & Wolf 2016
Ecology <https://esajournals.onlinelibrary.wiley.com/doi/full/10.1002/ecy.1484> for
examples. If the authors choose to retain these analyses, justification for using
hypervolumes or other aggregate measures of multi-dimensional trait variability in
relation to competition ought to be mentioned in the introduction.

**Response:** Thanks for your insightful comment. It is indeed an open question whether
one should use a multidimensional or one-dimensional trait approach to study ITV
responses to competition (Bennett *et al.* 2016; Bittebiere *et al.* 2019; Carmona *et al.*
2019; Funk & Wolf 2016; Kraft *et al.* 2015; Laughlin 2014). For example, Funk and
Wolf (2016) elucidated the importance of individual traits such as high specific root
length, low root-shoot ratio, and low leaf nitrogen content in determining a species'
ability to compete at high levels, while Kraft (2015) found that individual trait was more
highly correlated with fitness differences between species, and combinations of
multidimensional traits were more highly correlated with niche differences, which
provides experimental support for species coexistence in multiple dimensions and
challenges the simple use of one-dimensional traits to infer community assembly
processes. We argue that both one- and multidimensional trait approaches have their
unique strengths. We thus assessed the experimental results using both
multidimensional and unidimensional measures. Our results show that both approaches
yielded consistent results, suggesting that the choice of methods has little impact on our
main conclusions.

We have added the reasons why we used both multidimensional and one-
dimensional measures in Methods section (lines 505-510), and also briefed the two
approaches in Introduction (lines 113-115).

21. Line 309: How and why are results combined from the heterogeneous environments?
Do species vary in traits that allow them to be more or less successful in these different
treatments? If yes, could this bias results?

**Response:** To test whether the trade-off persists in spatially heterogeneous
environments, conspecific seedlings across abiotic conditions were combined as a
population, as conspecifics of a species in nature often exist in spatially heterogeneous
habitats. We have clarified the methodology in lines 414-417.

If we understand the second question correctly, yes, species varying in traits make
them differentially successful across distinct environments. Aggregating conspecific
across environments should not bias but rather provide more realistic estimations of
ITV and competitiveness, as shown in previous studies (Allesina & Levine 2011),
because conspecifics in nature often experience heterogeneous environments.

22. Lines 466 – 478: I'm wary of the discussion related to the directionality of changes
in traits since the results presented do not mention directionality of trait changes, just
changes in IV. The discussion ought to more directly relate to the results presented.

**Response:** We have removed the discussion related to directional trait changes and
focused on the changing patterns of ITV and their possible drivers (lines 290-301) in
the revised ms.

23. Lines 487 – 492: Given that competition intensity does not necessarily translate to
long-term coexistence, I think this conclusion is a bit of a stretch here and throughout
the manuscript.

**Response:** We have toned down the connection between our results and species
coexistence in our conclusion (lines 330-332) and as well as other sections of the text
(lines 315-319).

**Reviewer #2 (Remarks to the Author):**

Overall comments

1. This manuscript seeks to clarify how intraspecific trait variation can mediate
competitive species interactions (and coexistence), and how IV might vary in response
to the environment. This is an important area of research that is largely overlooked, and
the authors use an extensive dataset of seedlings to address their questions. I struggled
to follow the paper conceptually, primarily due to the vagueness of the text and the lack
of clear examples. The main hypothesis, that inferior species should have more IV than
superior species, is not an intuitive one, and the authors should explain the mechanism
in more detail as it would seem that the amount of variation should depend on the trait
in question.

For example, when there is strong competition for light, I would expect strong
directional selection for taller stature in the superior species (and possibly less IV),
whereas I would expect no influence on IV in height among the inferior species. The
inferior species may instead grow laterally, or make thicker leaves with more
photosynthetic tissue, or alter the leaf angle, such that there is no impact on height. I
was looking for more explanation and examples of this sort within the text. But it seems
as though it was written in a very broad way so as to describe trait variation in general,
which I don't think worked well.

**Response:** Thank you for your constructive feedback that is very helpful in improving
our study. Sorry for the vagueness of the original text regarding the trade-off hypothesis.
In the revised manuscript, we have clarified the trade-off and the underlying
mechanisms in greater detail with clearer examples (lines 79-87).

It is probably true that not every trait shows the trade-off. We have expanded the
discussion of this caveat in Methods (lines 505-510 and 535-537) and Discussion (lines
302-322). However, we did find that ITV decreased in many superior species (Table
S2) and have discussed this pattern on lines 246-258.

2. Apart from the conceptual issues, I had many questions about the methods and the
experimental design, which was also lacking some detail. I thought the figures were
very well made as were the tables, but again, I really struggled with the conceptual side
of this paper, and I wasn't fully convinced that IV should be higher in the inferior
species, at least not without more examples from their own study and from previous
studies. The one section of the paper where they do provide examples using below- and
aboveground traits is also incorrect in that they incorrectly assert that higher root tissue
density means a more acquisitive strategy, when it is actually the opposite. And there's
one short sentence that notes that the patterns of variation in IV will depend on the trait
in question.

**Response:** After submitted, we realized the method section needed much improvement
to make the experimental designs and methods understandable to the general reader. In
the revision, we took great effort to revise the method section (see sections 'Study site
and species selection' and 'Experimental design'). The change is so substantial that the
section has been expanded from 4 pages to 6 pages. We also recognized the need for
more empirical evidence of the interspecific mean-variance trade-off. Unfortunately,

we were unable to find any study that directly tests the trade-off, because no prior work
simultaneously quantifies both ITV and competitiveness between competing species.
We have acknowledged this limitation and highlighted the novelty of our experiment
in addressing this research gap. That said, we do find two types of indirect evidence
that suggest ITV may be higher in inferior species. First, the ITV might increase under
competition through plastic trait responses. For example, Gruntman *et al.* (2017) found
the clonal plant *Potentilla reptans* displayed plastic responses like ‘confrontational’
vertical growth, shade tolerance, and lateral-avoidance when competing with neighbors
varying heights and densities. These responses often vary substantially in direction and
magnitude among conspecifics (Novoplansky 2009). This potentially increase the ITV
of the clonal plant. Indeed, Bittebiere *et al.* (2019) found competition led to greater ITV
(using a multidimensional trait space based on ramet and connect traits) of two clonal
plants. Second, inferior species may have greater trait plasticity sensitivity to
competition. Bennett *et al.* (2016) found after 6 months of competition, the magnitude
of specific leaf area (SLA) plasticity in herbs rose proportionally with the competitive
suppression. This implies inferior species experiencing more competitive suppression
would undergo larger competition-induced trait changes. Longuetaud *et al.* (2013) also
showed that inferior *Quercus* species had higher crown plasticity, while superior *Fagus*
*sylvatica* exhibited lower plasticity in Western European mixed forests. Combining
these two types of evidence leads us to expect that inferior species have larger ITV than
superior species. We have reviewed these studies on lines 76-90.

Furthermore, our results directly support these expectations: inferior species in the
competition treatment exhibited a trend towards greater ITV for most individual traits
(Fig. S6). Specifically, root traits such as root tissue density (RTD, slope = -0.51, $P=$
0.018) and specific root length (SRL, slope = -0.46, $P = 0.023$) both were significantly
and negatively correlated with the species competitiveness. In other words, the greater
the competitive pressure on the species, the greater ITV in these root traits. These
findings are included in our revised manuscript (lines 290-301).

In addition, we agree the connection between ITV and species competitiveness may
vary across individual traits. For instance, leaf toughness (LTO) in inferior species
displayed lower conspecific variability versus superior species (Fig S6a, b). We also
observed a positive relationship between conspecific variability of LTO and species
competitiveness (Fig. S6). Interspecific competition even contributed to 2.74%
decreased LTO variability (Fig. 4). These observations align with the notion that plants
may adopt specific trait strategies, such as lower LTO, to enhance resource utilization
efficiency and photosynthesis, ultimately supporting faster growth (Kitajima & Poorter
2010). We have expanded the discussion about how conspecific variability differs
among individual traits on lines 302-314.

364 **Line by line edits**

3. Line 35 How is an inferior or superior species defined? I did go to the appendix
eventually but it seems it should be better defined early on.

**Response:** Thanks for the helpful suggestion. We have provided a concise explanation
in lines 84-85 and 113-115.

4. Line 37: how is a competitive response evaluated? In terms of a reduction in plant
size? For example?

**Response:** We realized that "competitive response" was ambiguous in this context and
have replaced it with the more precise term "plastic response" (line 27), which was
quantified as the change in the species' ITV from competition-free to competition
treatments.

5. Line 39: is it a greater increase in IV or the amount of IV itself? If it's the former,
then that's slightly different from the theory and perhaps needs clarification.

**Response:** Thank you for catching this typo - it is indeed the amount of ITV. We have
corrected it in lines 24-25. We also found inferior species exhibited a greater increase
in ITV when comparing between competition-free and competitive environments (Fig.
2). We also have clarified this point (lines 30-32).

6. Line 43: incomplete sentence

**Response:** Revised (lines 32-34).

7. Line 63: this sentence is a bit unclear. When two species have the same IV, the
superior species is defined as that which benefits more from having a wider trait space
(or possibly ecological niche?). I don't understand how the increased per capita seed
production comes into play. Because a wider trait space would mean that some
individuals have lower trait values and some have higher trait values, in other words
often a greater spread of values. I suppose the key point here is that it's not just about
the mean trait values but also the spread around those values that determines the
strength and direction of the interaction. I think this paragraph could be clearer.

**Response:** Apologies for the vague sentence. What we meant to explain is why the
relative magnitude of ITV between species is critical for competition outcomes. In Hart
et al.'s (2016), superior species were characterized by low competitive sensitivity, and
*per capita* seed production depends nonlinearly on competitive sensitivity in Hart's
model. So, variation, not just the mean, of the competitive sensitivity among
conspecifics will change the population's mean of *per capita* seed production based on
Jensen's inequality (Jensen 1906) and therefore the realized population growth rate and
competition outcomes (Bjørnstad & Hansen 1994). Hart *et al.* (2016) found that
incorporating equal ITV into competitive sensitivity of competing species reduces
coexistence, mainly because superior species benefit more in fitness than inferior
species from increased ITV. ITV promotes coexistence only when inferior species
exhibit much higher ITV, because the greater ITV in inferior species causes some
individuals with equal or lower competitive sensitivity versus superior species. This
bolsters the realized population growth rate of inferior species and enables species
coexistence.

Above and previous explanation involves too much model detail and obscures
mathematical theories like Jensen's inequality, marring the understanding of general

readers. In the revised manuscript, we provide a simpler explanation on lines 52-61 for
why the relative magnitude of ITV is important for species coexistence.

8. Line 78: between competing species or between species in general? It's an important
distinction as far as theory.

**Response:** It is the relative magnitude of ITV between competing species, because
competitiveness rankings between species can vary in different competition contexts.
For instance, in our pairwise competition treatment, *Quercus chenii* was inferior species
in *Quercus chenii* - *Hovenia acerba* pairs but became superior in *Cyclobalanopsis*
*myrsinifolia* - *Quercus chenii* pairs. Thus, we classified species as inferior or superior
based on the observed competitive intensities they experienced. We then examined the
relative ITV magnitude between these real competing species.

9. Line 79: suppression in terms of plant size? Survival? Reproduction? I just want to
know how this is defined.

**Response:** As in other studies (Aschehoug & Callaway 2014; Bennett *et al.* 2013), we
referred the term "suppression" to the reduction of seedling biomass due to the presence
of competing species. We added detailed information in the revision in lines 113-115.

10. Line 90: I didn't follow this sentence; can you explain further?

**Response:** Sorry for the vague sentence. Specifically, we intended to argue that the
magnitude of ITV is not simply linearly correlated with the width of species' resource
niche. Species with greater ITV do not necessarily have wider resource niche, and vice
versa. This lack of a one-to-one relationship makes it difficult to predict the relative
magnitude of ITV between competing species from a resource niche perspective alone.

In this revision, we realized the above explanation added unnecessary complexity
when introducing the expected relationships between ITV and species competitiveness.
Therefore, we have removed the discussion about the relationship between ITV and
resource niche width and simplified the sentence in lines 73-74 to make it more concise.

11. Line 95: unclear why 3 years was used as an example. What trait was measured in
these example studies that are referenced? And some species don't even live 3 years so
again, the example seems arbitrarily chosen. Some traits change within seconds to
minutes.

**Response:** We agree that experiments on trait change depend on the selected traits and
they can change quickly. We thus removed the temporal information that is not
contextualized here and added the corresponding temporal information in subsequent
examples (lines 81 and 82).

12. Line 98: unclear what is meant by different changes in IV between competing
species. Do you mean the sensitivity of IV could vary across species? That's my
interpretation based on the following sentence.

**Response:** Yes, it is the different sensitivities of ITV to competition between species.
We have made this clear in lines 76-79.

13. Line 101: I don't see how one results from the other. I don't follow the logic here.
Just because a species is an inferior competitor doesn't mean it's going to experience
more (or less) trait variation. More explanation is needed to substantiate this argument.

**Response:** We have expanded and refined our reasoning for expecting inferior species
to exhibit greater ITV in lines 79-87.

14. Line 103: vague

**Response:** We state here that the main reason that competition increases ITV of
competing species (especially inferior species) is that conspecifics of that species have
different directions and magnitudes of trait changes in the competitive environments.
We have rewritten this sentence to make it clearer (lines 76-87).

15. Line 107: can you provide an example? For example, plant height is a trait that
would vary in response to increasing abundance/density of neighbouring plants
(whether the same species or different species). But height wouldn't necessarily
become more variable, if there is strong directional selection to become taller in
response to more competition.

**Response:** Thanks for your insight. While plant height would serve as a useful example
for illustrating variable plastic responses to competition, it was not measured in our
study. As a result, we were unable to provide empirical evidence for height variability
in the subsequent Results and Discussion sections. Instead, we used stem-specific
density (SSD), which was measured in our study, as a representative example to
illustrate the concept. Specifically, taller plants often adopt rapid vertical elongation
strategy to maximize light access above the canopy (Gruntman *et al.* 2017) and usually
became superior species at seedling stage. This directional selection can result in lower
SSD for superior species, potentially reducing SSD variability for superior species.
Indeed, our study observed 2 of 8 superior species had reduced SSD variability (Fig.
S7a, d). Moreover, we also observed 4 of 8 superiors showed reduced multidimensional
ITV from competition-free to pairwise competition (Table S3). We have provided
discussions and specific examples of different ITV responses in superior species on
lines 246-258 in order to simplify the logic flow of the Introduction and allow for a
fuller discussion of this trait-specific pattern later on.

16. Line 144: please add the country

**Response:** Done.

17. Line 163: approximately how large were the seedlings at the time of transplant? I
think the seedling size at the beginning should have a strong impact on future
competitive ability. Were these species reared in high light, high moisture, high
nutrients? There's almost no information about the environmental conditions the plants
experienced.

**Response:** Sorry for missing the information on the initial seedling height and nursery
conditions. Seedling heights at transplantation were approximately 6-25 cm, with

deciduous species averaging 12-25 cm and evergreen species averaging 6-13 cm. We
listed seedling initial heights for each species in Table S5 and lines 364-366. Following
your suggestion, we constructed a mixed-effects model to explore the relationship
between first-measured height (two months after germination) and competitiveness for
all seedlings in the second phase of the experiment, with species as a random effect.
Our findings revealed a significantly positive correlation ($Pseudo R^2 = 0.366$, $Slope =$
0.018 , $P < 0.001$), indicating that taller seedlings were more competitive (higher *RII*
values). We have now included this information in the revision (lines 352-353).

In nursery period, adequate sunlight and water were provided during the seedling
germination stage. The soil used for rearing was topsoil (15-30 cm) collected from the
forest containing the mother trees, with 0.61 g/kg total carbon, 0.04 g/kg total nitrogen,
0.45 g/kg total phosphorus and a pH value of 5.21. We have added this information on
lines 356-362.

18. Line 168: how far apart were the seedlings? If the seedlings were quite small, they
would have weak competition but if one was larger than the other, it would be
asymmetric. Note: this information is presented on line 205 and should definitely be
moved up sooner.

**Response:** A 3-5 cm distance was used to foster competition within the pot, along with
the 15-20 cm pot spacing to minimize between-pot interference. Additionally, to ensure
fair comparisons, the initial size of the conspecific seedlings was kept consistent for
each species among pots, as indicated in Table S5. We have moved the above
information forward from Phase II to Phase I in Methods (lines 391-394). There were
inherent differences in the initial heights between the species, likely resulting in
asymmetric competition. We have now included this information earlier in the
manuscript (lines 364-368).

19. Line 181: insufficient detail regarding the environmental conditions. What was the
mean and maximum for each environmental property? Providing the minimum light
doesn't tell us much about potential light stress. I see that an appendix is referenced but
I don't think we should need to see the reference for this very basic information about
growth conditions. Even just a few sentences would be useful.

**Response:** Thank you. A brief introduction about the environmental conditions was
added in Methods (lines 419-445).

20. Line 196: can we also know the range of variation in the environmental variables
over the course of a day or months? The term homogeneous is somewhat subjective as
it depends on the time frame.

**Response:** Homogeneity in this study refers primarily to spatial consistency of abiotic
environment across pots within a given environmental block, rather than temporal
invariance. Abiotic factors like light intensity naturally vary over time. We have
redefined homogeneous environment in more detail in lines 399-409 and 413-414.

21. Line 222: it would have been helpful to know earlier that competitive effects were
based on biomass

**Response:** We now state in the Introduction (lines 113-115) that competitive ability is
based on seedling biomass in our study.

22. Line 225 and line 236: the authors say this is to avoid confusion yet it is very
confusing. The definition of R_{II} is defined in terms of biomass when a species is within
a pair, yet this line says that you use the term competitiveness even when the species is
not in a pair. So does this mean that when I see the terms superior or inferior, that this
species may not necessarily be in a pair? That it could be on its own? Because then later
you note how the characterization of a species as being superior or inferior depends on
the identity of the other species within the pair.

**Response:** Sorry for the confusion. For a given species group (species i or species j), if
species i has greater value of R_{II} when competing with species j , then species i and j
are called superior and inferior species for that species group, respectively. These
competitiveness labels are also retained for that species group in the competition-free
environment. This design simplifies comparing ITVs between species in that group
across competition and competition-free treatments.

It's important to note that a species can be either competitively superior or inferior
in different groups of species, as competitive ability is not a constant attribute of a
species but can change depending on whom the species compete with and by resource
availability. This is particularly relevant in our study, as Phase I includes eight different
groups of species, and a species may be included in more than one species group. We
revised these text in lines 468-479.

23. Line 279: is it possible to also calculate the amount of variance around the mean
hypervolume? Based on Figure 2 it looks like you have the standard error

**Response:** Yes, we have calculated the mean and standard error of hypervolume for
each species. See detailed methods in lines 526-529.

24. Line 230: and by resource availability

**Response:** Revised (line 475).

25. Line 407: I disagree that this is the case for all traits. That's my main argument.
That some traits will tend to be quite variable within species even in the absence of
competition or heterogeneity. For example, leaf size is hugely variable in some species
due to leaf ontogeny.

**Response:** We agree that not all traits exhibit low variability in the homogeneous,
competition-free environment. For example, in our study, the chlorophyll content (Chl)
of seedlings under competition-free homogeneous environment showed higher ITV
(1.18 in *Bao's CV*) than the mean ITV (1.16 in *Bao's CV*) under competitive
homogeneous environment. Therefore, what we intended to convey is that *most* traits
have lower ITV in a competition-free homogeneous environment compared to

heterogeneous or competitive environments. To accurately express the above meaning,
we have added that most traits are consistent with the overall pattern (lines 216-217).

26. Line 418: optimal being defined in terms of max biomass?

**Response:** Generally, ‘optimal’ is defined in terms of over-all performance, such as
maximum biomass and seed production, as well as low death rate. In our 3-years study,
most seedlings were still alive and far from mature to produce seeds, so for our study,
‘optimal’ just refers to maximum biomass. We have added the above information in
lines 216-221.

27. Line 418: a concave down trait performance curve but for which trait(s)? I don’t
see how you can generalize so broadly.

**Response:** We now agree that the shape of trait-performance curve likely varies for
different traits and environments (Li *et al.* 2022; Siefert & Laughlin 2023; Stears *et al.*
2022). Thus, it is an overgeneralization to apply theoretical trait-performance curves
(Hart *et al.* 2016; Stump *et al.* 2022) in our experimental study. Furthermore, we
realized these curve discussions do not actually explain the differential ITV changes
between competing species. So we removed that content and instead added text on
potential mechanisms driving the observed distinct ITV changes between competing
species in lines 229-245.

28. Line 420: why would an individual produce a suboptimal trait value in response to
a cue? Are you suggesting that an inferior species in low moisture, might have some
individuals that make deeper roots and others that make shallower roots, even if the
more shallow-rooted individuals perish? Is this part of the reason why these species are
inferior in the first place? Are you suggesting this is maladaptive? It seems that there
should be more discussion about alternative resource use strategies and performance
landscapes (*sensu* Laughlin). A concave down trait performance curve versus a concave
down doesn’t tell us anything about the extent of variation though. They could have
different shapes but the exact same amount of variation, so the sentence that follows
from this (Line 424) doesn’t exactly follow the logic (in my opinion).

**Response:** Thanks for the thoughtful comment that inspired us to think more deeply
why individuals of superior species display suboptimal trait values under competition.
Upon reflection, we realized our previous discussions about suboptimal trait values and
trait-performance curves had some limitations. We therefore removed those texts and
added new explanations covering possible drivers behind the distinct ITV changes
between competing species in lines 229-245.

29. Line 469: negative or positive change? Add + or –

**Response:** It is positive change. Revised accordingly (line 291).

30. Line 475: it does not make sense to me why a species would be conservative above
and acquisitive below. I think the use of those terms is not helpful in this context. Higher
LDMC could be associated with longer leaf lifespan, and higher stem density usually

helps to ameliorate water stress, so the higher root tissue density should be a drought
adaptive response, not evidence of a more resource acquisitive strategy. Root tissue
density also typically increases with a reduction in soil nutrients, so again this isn't
evidence of being more acquisitive on bottom and conservative on top.
**Response:** Thank you for the valuable comment. We have removed the discussion
about conservative and acquisitive strategies and instead focused on adaptive responses
of traits in lines 290-314.

**Reviewer #3 (Remarks to the Author):**

This study explores differences in intraspecific variability between competitively
inferior vs superior tree species in order to test for an interspecific mean-variance trade-
off. The authors use an experimental approach that clearly involved a massive amount
of work and is impressive in its scope. The results provide novel experiment evidence
of the mean-variance trade-off, which is relevant for species coexistence. Overall, the
study makes a nice contribution to our understanding of the role of intraspecific
variation in structuring diverse communities.

**Response:** Thank you for the encouraging feedback.

2. My main concern about the study is that the authors provide no information about
where or how the seeds used in the experiment were collected (e.g., over what spatial
extent, how many source trees were used, etc). This information is critical for any study
of intraspecific variability (IV). If competitively inferior species were rarer in the
community and/or produced fewer seeds than competitively superior species, then there
might be the need to collect seeds of inferior species from more source trees and over
a larger spatial extent, which could bias estimates of IV.

**Response:** This is an important point. We have added detailed information about seed
collection in lines 351-353. In brief, seeds for each species were meticulously collected
from one healthy, mature parent tree within a 1 km² area of our study region. This
ensured the low genetic variation among conspecific seeds and minimized maternal
effect differences among species. However, it suffers a common experimental
limitation by reducing representation of our ITV for broadly distributed species. We
acknowledge this limitation in lines 352-353.

3. On a related note, the authors appear to be using the term IV to mainly refer to
phenotypic plasticity. However, IV can arise due to genetic variation and phenotypic
plasticity (and there can also be genetic variation in phenotypic plasticity). For the
authors' first question about IV in a competition-free, homogenous environment, then
the IV would be the result of genetic variation (since you would not expect to see
plasticity under uniform conditions). For questions 2 and 3, about IV under competition
and in heterogenous abiotic environments, then they are talking about phenotypic
plasticity. The authors need to be clearer about the underlying drivers of IV (eg genetic
variation and phenotypic plasticity). This is related to my comment above about seed
collection, since that will influence how much genetic variation exists in the seed pool
used in the experiment as well as how plastic the collected individuals are expected to
be (eg if some species were collected from more heterogenous environments than other
species).

**Response:** We greatly appreciate the suggestion to clarify the mechanisms underlying
ITV. In the original draft, we aimed to elucidate this. However, our experimental design
cannot precisely distinguish ITV sources, as neither genetic variation nor phenotypic
plasticity was fully controlled in our study. In fact, controlling the drivers of ITV
remains challenging for trees. Thus, we have narrowed our study scope by removing
the discussion on adaptation and evolution from the introduction and just focused on

our core questions and findings. Meanwhile, in the discussion section (lines 220-226),
we thoroughly considered potential mechanisms producing the observed trade-off
given our experimental design.

4. My other concern is about the influence of growth form on the results. Specifically,
the two deciduous species included in the experiment were the most competitively
superior in both Phase 1 and Phase 2 experiments. Deciduous species employ a different
ecological strategy than evergreen species, which typically means that they differ in
other traits as well. Do the observed relationships remain qualitatively similar if
removing those two species from the analyses? (I don't mean whether they are still
statistically significant because they probably won't be just due to the smaller sample
size, but rather whether the trend is the same.)

**Response:** This is a very constructive comment which would strengthen the robustness
of our study. Following this comment, we conducted an additional analysis by
excluding deciduous species. The results (Fig. S5) remained qualitatively similar.
Specifically, inferior species in multispecies competition treatments still exhibited
greater ITV, as shown by the significant negative slopes of -7.57 (Fig. S5e) and -24.39
(Fig. S5f), respectively. Analyzing competition-free heterogeneous environment
showed a similar pattern, with a significant slope of -15.42 (Fig. S5c). In the pairwise
competition under homogeneous environment, the number of species pairs dropped
from 8 to 3 without deciduous species. While the overall pattern was not significant due
to low sample size exactly as you suspected, a weaker decreasing trend was observed
(Fig. S5d). These results together support the presence of the interspecific mean-
variance trade-off in our study. We have included this information in Methods (lines
561-563) and Results (Lines 163-164).

**New Fig. S5** | Relationships between intraspecific trait variability (mean \pm standard
 error, SD^7 , cube-root transformation) and competitiveness of evergreen species (points
 under competition-free (hollow points, a-c), pairwise competition in homogeneous
 environment (orange circles, d, $N=3$) and multispecies competition in both
 homogeneous (blue squares, e, $N=5$) and heterogeneous (purple triangles, f, $N=5$)
 environments. In a and d, the gray circles represent inferior and superior species in the
 3 evergreen species pairs, and the orange circles represent their mean values.
 Intraspecific trait variability was quantified by the hypervolume of 7 traits. The gray
 and orange lines represent the pairwise competition relationship and other solid ($P <$
 0.05) and dashed ($P > 0.05$) lines are simple linear regression lines with a 95%
 confidence interval. NS means non-significant and black asterisks indicate significant
 levels (\bullet : <0.1 ; $*$: <0.05 ; $**$: <0.01 ; $***$: <0.001).

**Some additional comments:**

5. Line 65: Why just “per capita seed production”? Wouldn’t any increase in fitness
 have this effect?

**Response:** Our mention of “per capita seed production” specifically referred to the
 metric discussed in Hart *et al.* (2016) study we cited. We have revised the sentence to
 reflect broader potential fitness impacts from ITV (lines 52-56).

6. Line 69: add “the” before “same”

**Response:** Done (line 60).

7. Line 71: add “the” before “Advantage”
**Response:** This word was removed in the revised manuscript for brevity.

8. Line 104: It is not clear here why this would lead to large IV in only inferior species
(and not also in superior species)
**Response:** We have now provided an explanation (lines 79-87).

9. Line 106-107: I don’t follow the logic here – why would it deviate from its optimal
trait value. If IV is increased to enhance a species advantage, presumably it is achieving
the optimal trait value for a given environment.
**Response:** It indeed does not play out well in logic. To straighten out the logic flow in
the Introduction, we have removed these texts about suboptimal trait values. Instead,
we added a more comprehensive explanation of why superior species can display
distinct ITV changes in lines 249-253.

10. Line 126: do these 10 tree species coexist in nature? Are they all native?
**Response:** They are native species and grow in the study area. We have added this
information in Line 109.

11. Line 144: indicate the country where the site occurs (China)
**Response:** Done.

12. Line 149: What was the rationale for selecting both evergreen and deciduous (and
11 of one type but only 5 of the other type)? Was there some expectation about how
they would differ?
**Response:** The 16 dominant and codominant native tree species in our study area were
chosen based on previous knowledge about species composition in the study area while
constrained by seed availability. Specifically, forests in our study area are subtropical
evergreen broad-leaved forests with noticeable deciduous elements (Yang *et al.* 2010).
Thus, we selected 11 evergreen and 5 deciduous tree species common in our study area
(Wang *et al.* 2020; Yang *et al.* 2021). Furthermore, only those species with large mother
trees within a 1 km² area were included to ensure adequate seed sourcing (lines 350-
353).
We expected that evergreen tree species to have greater ITV if the trade-off exists
in our forest, because our previous experiment revealed deciduous tree species often
exhibit faster growth and superior seedling competitiveness (Yang *et al.* 2021). We
have added these considerations in Methods (lines 344-351 and 364-368).

13. Line 151: As mentioned above, there needs to be a lot more detail about seed
collection (how many mother trees, over what geographic area and types of
environments, were all species collected at the same sites, etc).
**Response:** Thank you for this question about seed sourcing. In brief, we collected seeds
of each species from a mature, healthy adult tree within our study area. We have added
detailed description about seed collection in Method (lines 349-353).

14. Line 158: What kind of homogenous environment? What resources were the species
likely competing for in that environment (e.g., where soil nutrients limited, was there
limited light or water, etc)?

**Response:** The term "homogenous environment" in this study refers to consistent
abiotic conditions both within and between pots in the same environment treatment. To
ensure this spatial homogeneity, 3.5 kg of thoroughly mixed soil collected from the
study area was added to each pot. Additionally, light intensity was kept consistent by
using the same black netting above all pots, with monthly pot rotation to minimize
potential micro-variation in light exposure.

The most limited resources in our experiment were different among environmental
treatments. In terms of resource competition, our study primarily focused on
aboveground light resources and belowground water and nutrient resources. Different
environmental treatments may vary in resource availability. For instance, under low
light conditions in certain blocks, plants may intensify competition for light, while
under low phosphorus or limited water conditions, they may compete more intensely
for soil nutrients or water, respectively. It is worth noting that plants may face multiple
constraints from the aforementioned abiotic factors simultaneously.

We have provided more detailed information about the homogeneous environment
and limited resources in the Methods (lines 399-409; lines 420-430).

15. Line 180: It is not clear here or anywhere in the main text or supplement how the 9
different environments were selected. It was not all combinations of the 3 variables, so
why were some combinations excluded?

**Response:** Testing all 27 potential combinations of the 3 environmental factors at 3
levels each, with the same 40 replications per combination in our current experiment,
would have required 9,720 pots and 17,010 seedlings in total, far exceeding our
experimental capacity. To balance feasibility and representation, we selected a subset
of 9 environments using a 3x3 orthogonal design table, enabling the exploration of
environment combinations with available resources. We have added this rationale for
our environment selection approach in Methods, lines 414-430.

16. Line 198: "populations of two competing species" is confusing here because
competition was not pairwise in Phase II

**Response:** Phase II contains multiple species, not two. We have rewritten the sentence
and moved "populations of two competing species" to lines 414-417 in the revised
manuscript to improve clarity.

17. Line 277: Was this done separately for competition-free and competition treatments?

**Response:** Yes, we have revised this sentence accordingly (lines 525-526).

18. Line 279: The hypervolume values were averaged separately for each species under
each scenario (e.g. competition-free homogenous, competition-free heterogenous, etc)?

**Response:** You are right that hypervolume was calculated per species under each
scenario. We made it clear in lines 526-529.

19. Line 280: State how many individuals had missing trait data

**Response:** Before quantifying species' hypervolumes, we excluded 74 and 543
individuals with missing trait values in Phase I and in Phase II, respectively. We have
added this information in Methods (lines 396-398 and 456-457).

20. Line 288: was the change in IV calculated in absolute or relative (ie % difference)
terms? Looking at the graphs, it appears to be absolute differences. Do the results
change if using % difference instead?

**Response:** Both absolute and relative ITV changes were calculated when comparing
competition-free to competition treatments and they exhibit consistent trends.
Specifically, the y-axis in Fig. 4 represents the absolute per-trait change in ITV. The
numbers displayed on the right side of each point indicate the relative ITV change. We
have clarified this in lines 538-541 and 565-568.

21. Line 292-294: Why was this only done for Phase II?

**Response:** In Phase II we deliberately selected 7 of 10 traits to calculate ITV, while
Phase I used all traits. This selection aimed to make the multidimensional trait space
comparable between phases. To verify the selection of traits did not bias Phase II, we
added analyses using all traits. We have clarified this point in lines 543-547.

22. Line 303-305: Why wasn't species included as a random effect? It sounds like you
have multiple values for each species in the model (eg one for each treatment), but are
not accounting for the fact that values from the same species are not independent of
each other.

**Response:** Thank you for the suggestion. We believe by non-independence, you mean
that the same species is not independent between the two competition treatments? In
this case, we have made revision to clarify the modeling approach in Methods (lines
552-561). Specifically, we constructed separate mixed-effects models for competition-
free and competition treatments, each with a random intercept and slope. Because they
were modeled separately, the non-independence of trait values among conspecifics
could not bias our results.

Alternatively, if non-independence means dependence of species across the 9
environmental blocks, we need to build new models with both the species and
environmental blocks as two cross-over random factors. Results (Fig. S4) were
consistent with our previous findings, i.e., it still shows significant negative
relationships between species competitiveness and ITV in 7-dimensional (Slope = -3.68,
$P = 0.033$, Fig. S4b) and reduced-dimensional trait space (Slope = -1.55, $P = 0.017$, Fig.
S4d) in a homogeneous multispecies competition environment. We have added the
above information to Methods (lines 558-561) and Results (lines 161-163).

**New Fig. S4** | Overall relationships between intraspecific trait variability (mean \pm
 standard error, cube-root transformation) and competitiveness of species (points) under
 competition-free (colored hollow points, a, c) and multispecies competition (colored
 hollow points, b, d) in 9 homogeneous environments. The top right panel shows the
 relationship between intraspecific trait variability and competitiveness within each
 abiotic environment. Intraspecific trait variability was quantified by the hypervolume
 of 7 traits. Intraspecific trait variability was quantified by hypervolume of the raw 7
 traits (a-b) and the reduced dimension first 3 principal axis traits (PC1-PC3, c-d),
 respectively. The solid ($P < 0.05$) and dashed ($P > 0.05$) lines are simple linear
 regression lines with a 95% confidence interval.

23. Line 308: Was this done using data from treatments with or without competition?

**Response:** The Wilcoxon rank-sum test was conducted separately for the competition
 and competition-free treatments. We have clarified it in lines 550-563 and 567-571.

24. Line 312: Using Phase I or Phase II data (or both)?

**Response:** Both. We have made it clear in lines 568.

25. Line 310-314: Why didn't you model IV as a function of competition, RII, and RII
 x competition interaction (with species as a random effect)? That would be a more
 straightforward test of 1) whether competition affects IV and 2) whether the
 relationship between RII and IV changes under competition. If you model it that way,
 are the results consistent?

**Response:** Thank you for the very helpful comment. Following your suggestion, we
 employed new mixed-effects models assessing the relationships between competitive
 treatments (competition-free and competition), competitiveness (*RII*), and their
 interaction on species' ITV. We ran these analyses separately for homogeneous and
 heterogeneous environments, with species as a random factor. The new results (Table
 S1) were consistent with our previous findings in Fig. 2b-c and e-f. We have
 incorporated these new models and results in pages 26-27 and 7-9). However,
 interpreting results of models with a binary variable and an interaction of binary and
 continuous variables is not straightforward and needs advanced statistical knowledge.
 To enhance readability for a broader audience and obtain additional insights (e.g., R^2
 for each competition treatment), we also retained previous visualizations like Fig. 2 to
 intuitively examine the interspecific mean-variance trade-offs for the different
 competition treatments.

 **New Table S1** Summary of the mixed-effects models employed to explore the
 relationships involving competitive treatments (competition-free and competition,
 abbreviated as 'CompTreat'), competitiveness (*RII*), and their interaction concerning
 species' ITV within the context of multispecies experiment. These analyses were
 separately conducted in both homogeneous (specifically, environmental block 1, which
 corresponds to the Phase I environment) and heterogeneous environments, with species
 as a random factor. Values in bold indicate statistically significant effects ($P < 0.05$).

	Predictors	Estimates	CI	P
Multispecies experiment in a homogeneous environment	CompTreat_competition-free	0.67	-2.08 – 3.41	0.591
	CompTreat_competition	-0.37	-3.11 – 2.38	0.766
	competitiveness	-0.81	-5.43 – 3.81	0.696
	CompTreat_competition* competitiveness	-8.03	-14.56 – -1.49	0.022
			Pseudo R^2	0.879
Multispecies experiment in a heterogeneous environment	CompTreat_competition-free	-0.10	-3.35 – 3.16	0.947
	CompTreat_competition	0.32	-2.93 – 3.57	0.825
	competitiveness	-8.70	-14.42 – -2.98	0.008
	CompTreat_competition* competitiveness	-6.26	-11.99 – -0.54	0.036
			Pseudo R^2	0.900

26. Line 332: change “survival” to “surviving”

**Response:** Done (Line 143).
27. Line 337: specify that “significant suppression” is referring to reduced biomass.
**Response:** Revised (Lines 149-150).
28. Line 369-370: There is a typo here in the numbers – they do not match up with what
is on the graphs (and also how can you have an R-square value of -8.7).
**Response:** Corrected (lines 180-181). The negative value was the slope of the
regression line. We have adjusted our presentation to avoid such confusion.
29. Line 380: When it says “individual traits”, it makes it sound like you ran separate
models for each trait. But instead, it seems like this is just a single model for all traits.
This needs to be better explained both here and in the methods.
**Response:** Thanks! To clarify, our approach involved running both comprehensive
models that considered all traits collectively and separate models for each individual
trait. We have made this clear in the methods (lines 565-571) and the results (lines 188-
195, Fig. S6).
30. Line 382: Is this in the homogenous environment only? (Both here and throughout
the Results, make sure to specify whether you are referring to homogenous vs
heterogenous environment when referring to competition treatment results. It is often
not clear.)
**Response:** For simplicity, the result covers 3 competitive scenarios in our experiment
(corresponding to Fig. 2), including all seedlings in homogeneous two-species
competition, homogeneous multispecies competition and heterogeneous multi-species
competition. We have clarified these on lines 538-541.
31. Line 387: Specify “greatest” compared to what (ie other traits measured).
**Response:** It was compared with all other measured traits. The missing information has
been added in lines 197-199.
32. Line 398: change “seedlings stage” to “seedling stage”
**Response:** Done.
33. Line 407-409: How exactly are determining that the traits are optimal? It seems like
you are assuming that they are because of the low IV, but low IV does not necessarily
mean optimal trait values! Do you also see lower interspecific variation? The fact that
you find low IV in a homogenous environment just means that there is little genetic
variation in the trait in your seed pool, and that increased IV under heterogenous
environments and competition is resulting from phenotypic plasticity.
**Response:** We agree that the low species’ ITV in homogeneous environments is likely
due to low intraspecific genetic variation, and this has been added in lines 220-226.
Trait optimization is defined according to a narrowness of community trait
distributions. Specifically, if the distribution of traits in the plant community is in part

the result of selective pressure exerted by environmental conditions favoring species
with specific functional traits, the environment should then select traits that are optimal
or at least labile to the current environment, at which point the species should have low
interspecific variability and exhibit a narrow range of trait values. This is also known
as the trait optimization hypothesis (McGill *et al.* 2006). In our study, we found some
evidence supporting this hypothesis in homogeneous environments free of competition
because the ITV of most experimental species (Figs. 2a-b and S2a-b) and interspecific
trait dissimilarities of 62.5% species pairs (Table S6) were relatively lower in such a
simple environment compared to heterogeneous or competitive environments. We have
clarified and expanded this concept in the revised manuscript to provide a more
comprehensive explanation (lines 215-226).

34. Line 411-413: This makes no sense to me. If IV is selected for by the environment,
then presumably individuals would be accurately perceiving the environmental stresses
and adjusting trait values accordingly to increase fitness.

**Response:** We have deleted this sentence in the revision.

35. Line 414 and Line 430: specify what trade-off you are referring to

**Response:** It is the interspecific mean-variance trade-off. Clarified (lines 228 and 259).

36. Line 417-422: This section is confusing and hard to follow for anyone who is not
already very familiar with the papers being reference. Please rephrase to explain this
more clearly.

**Response:** Apologies for the confusing explanations relying on complex mathematic
models and theoretical assumptions about trait-performance curves. The complexity
and lack of empirical evidence led us to remove those previous discussions. Instead, we
have provided a new explanation with clearer logic and empirical evidence for the
drivers of interspecific mean-variance trade-offs in lines 229-258.

37. Line 431: remove the word “only”. Previous studies have also found that IV is
related to genetic variation, species life history strategies, dispersal modes, etc.

**Response:** Done (Line 261).

38. Line 460-461: rephrase as “which could allow invasive species to survive in the
face of...”. As written, it makes it sound like invasive species are actively choosing this
strategy, rather than high plasticity being one of the reasons they become invasive.

**Response:** Following the comment, we have revised the subordinate clause by
removing the active tone and improved clarity (lines 284-286).

39. Line 466: was this higher IV in root traits compared to leaf and stem traits found in
all environments? What about if you just look at environments that differ in light only?
I would expect above ground traits to response more to differences in light, and root
traits to respond more to differences in soil moisture and nutrients. Given that 2 of the
3 abiotic variables being manipulated in the experiment were below ground variables,

it is not surprising that root traits would appear to have higher IV. You need to consider
the specific abiotic variable when comparing the different traits.

**Response:** Thank you for the insightful feedback. We agree higher ITV of root traits
could stem from including more soil environmental variables. Unfortunately, because
our experiment was not a full factorial design (explained in lines 419-422), we were
unable to isolate the specific effect of individual environmental factor on each trait. We
added your point about higher root trait variation in Discussion (lines 297-301).

40. Figure 1: again, it is not clear how the 9 environments were chosen since it is not
all combinations of the 3 variables.

**Response:** Sorry for missing the information about the environment selection. We have
added this information into Methods (lines 399-409) and a brief explanation in the
figure caption (lines 801-804).

41. Figure 2 and Figure 3: Include exact P values for significant relationships, not just
$P < 0.05$

**Response:** Done.

42. Figure 2, line 692: Do you mean “orange circles, b-d” here? I think it is a typo
because only d has orange circles.

**Response:** Yes, it has been corrected.

43. Figure 3: something is off with the figure legends. For example panel b legend has
orange and blue lines for homogenous environment, but there are no orange and blue
lines in the graph. Also in panel A, it is not clear what the difference is between blue
and purple.

**Response:** Thanks. They have been corrected.

44. Figure 4: Is this from a homogenous environment only? Specify in the legend.

**Response:** No, this result covers 3 competitive scenarios in our experiment
(corresponding to Fig. 2), including all seedlings in homogeneous two-species
competition, homogeneous multispecies competition and heterogeneous multi-species
competition. Being clarified in both the legend and lines 538-541.

**References**

- Allesina, S. & Levine, J.M. (2011). A competitive network theory of species diversity.
*Proceedings of the National Academy of Sciences*, 108, 5638–5642.
- Aschehoug, E.T. & Callaway, R.M. (2014). Morphological variability in tree root
architecture indirectly affects coexistence among competitors in the understory.
*Ecology*, 95, 1731–1736.
- Banitz, T. (2019). Spatially structured intraspecific trait variation can foster
biodiversity in disturbed, heterogeneous environments. *Oikos*, 128, 1478–1491.
- Begon, M. & Wall, R. (1987). Individual variation and competitor coexistence: a model.
*Functional Ecology*, 1, 237–241.
- Bennett, J.A., Lamb, E.G., Hall, J.C., Cardinal-McTeague, W.M. & Cahill Jr., J.F.
(2013). Increased competition does not lead to increased phylogenetic
overdispersion in a native grassland. *Ecology Letters*, 16, 1168–1176.
- Bennett, J.A., Riibak, K., Tamme, R., Lewis, R.J. & Pärtel, M. (2016). The reciprocal
relationship between competition and intraspecific trait variation. *Journal of*
*Ecology*, 104, 1410–1420.
- Bittebiere, A.-K., Saiz, H. & Mony, C. (2019). New insights from multidimensional
trait space responses to competition in two clonal plant species. *Functional*
*Ecology*, 33, 297–307.
- Bjørnstad, O.N. & Hansen, T.F. (1994). Individual variation and population dynamics.
*Oikos*, 69, 167–171.
- Carmona, C.P., de Bello, F., Azcárate, F.M., Mason, N.W.H. & Peco, B. (2019). Trait
hierarchies and intraspecific variability drive competitive interactions in
Mediterranean annual plants. *Journal of Ecology*, 107, 2078–2089.
- Crawford, M., Jeltsch, F., May, F., Grimm, V. & Schlägel, U.E. (2019). Intraspecific
trait variation increases species diversity in a trait-based grassland model. *Oikos*,
128, 441–455.
- Des Roches, S., Post, D.M., Turley, N.E., Bailey, J.K., Hendry, A.P., Kinnison, M.T.,
*et al.* (2018). The ecological importance of intraspecific variation. *Nature*
*Ecology & Evolution*, 2, 57–64.
- Feniova, I.Yu., Aibulatov, D.N. & Zilitinkevich, N.S. (2013). Effects of individual
variability on the outcome of competition between cladoceran species. *Inland*
*Water Biology*, 6, 294–300.
- Funk, J.L. & Wolf, A.A. (2016). Testing the trait-based community framework: Do
functional traits predict competitive outcomes? *Ecology*, 97, 2206–2211.
- Gruntman, M., Groß, D., Májeková, M. & Tielbörger, K. (2017). Decision-making in
plants under competition. *Nature Communication*, 8, 2235.
- Hart, S.P., Schreiber, S.J. & Levine, J.M. (2016). How variation between individuals
affects species coexistence. *Ecology Letters*, 19, 825–838.
- Jensen, J.L.W.V. (1906). Sur les fonctions convexes et les inégalités entre les valeurs
moyennes. *Acta Mathematica*, 30, 175–193.
- Kitajima, K. & Poorter, L. (2010). Tissue-level leaf toughness, but not lamina thickness,
predicts sapling leaf lifespan and shade tolerance of tropical tree species. *New*
*Phytologist*, 186, 708–721.

Kraft, N.J.B., Godoy, O. & Levine, J.M. (2015). Plant functional traits and the
multidimensional nature of species coexistence. *Proceedings of the National*
*Academy of Sciences*, 112, 797–802.

Laughlin, D.C. (2014). The intrinsic dimensionality of plant traits and its relevance to
community assembly. *Journal of Ecology*, 102, 186–193.

Li, Y., Jiang, Y., Zhao, K., Chen, Y., Wei, W., Shipley, B., *et al.* (2022). Exploring
trait–performance relationships of tree seedlings along experimentally
manipulated light and water gradients. *Ecology*, 103, e3703.

Longuetaud, F., Piboule, A., Wernsdörfer, H. & Collet, C. (2013). Crown plasticity
reduces inter-tree competition in a mixed broadleaved forest. *European Journal*
*of Forest Research*, 132, 621–634.

McGill, B.J., Enquist, B.J., Weiher, E. & Westoby, M. (2006). Rebuilding community
ecology from functional traits. *Trends in Ecology & Evolution*, 21, 178–185.

Milles, A., Dammhahn, M. & Grimm, V. (2020). Intraspecific trait variation in
personality-related movement behavior promotes coexistence. *Oikos*, 129,
1441–1454.

Novoplansky, A. (2009). Picking battles wisely: plant behaviour under competition.
*Plant, Cell & Environment*, 32, 726–741.

Siefert, A. & Laughlin, D.C. (2023). Estimating the net effect of functional traits on
fitness across species and environments. *Methods in Ecology & Evolution*, 14,
1035–1048.

Stears, A.E., Adler, P.B., Blumenthal, D.M., Kray, J.A., Mueller, K.E., Ocheltree, T.W.,
*et al.* (2022). Water availability dictates how plant traits predict demographic
rates. *Ecology*, 103, e3799.

Stump, S.M., Song, C., Saavedra, S., Levine, J.M. & Vasseur, D.A. (2022).
Synthesizing the effects of individual-level variation on coexistence. *Ecological*
*Monographs*, 92, e01493.

Uchmański, J. (2021). Can a More Variable Species Win Interspecific Competition?
*Acta Biotheoretica*, 69, 591–628.

Uriarte, M. & Menge, D. (2018). Variation between individuals fosters regional species
coexistence. *Ecology Letters*, 21, 1496–1504.

Wang, M., Yang, J., Gao, H., Xu, W., Dong, M., Shen, G., *et al.* (2020). Interspecific
plant competition increases soil labile organic carbon and nitrogen contents.
*Forest Ecology & Management*, 462, 117991.

Wang, S. & Callaway, R.M. (2021). Plasticity in response to plant–plant interactions
and water availability. *Ecology*, 102, e03361.

Yang Q. *et al.* (2011). Community structure and species composition of an evergreen
broadleaved forest in Tiantong’s 20 ha dynamic plot, Zhejiang Province, eastern
China. *Biodiversity Science*, 19, 215–223.

Yang, J., Lu, J., Wang, R., Wang, X., Li, S. & Shen, G. (2021). Importance and benefit
of incorporating the responses of species mean trait values in trait-based
community assembly. *Ecological Indicators*, 130, 108095.

Reviewer #2 (Remarks to the Author):

General comments

This manuscript seeks to clarify how intraspecific trait variation can influence competitive interactions in superior (dominant) versus inferior species, considering resource availability. I appreciate that the authors have included much more information about the methods, and have tried to clarify the introduction and discussion in several places. However, I am still finding this paper hard to follow, unfortunately. I understand the big picture but the explanation of the details still needs work.

In reading the response to reviewer 1, I thought it was helpful to see a more formal definition of the mean-variance trade-off. However, I still don't think that the mechanistic explanation is clear enough for a broad audience that is unfamiliar with this work. Reading the main text (lines 52-63), it is still confusing, and written at a high level rather than being written at such a level that anyone could understand it. For example, the term competitive sensitivity is never defined, demographic rates are never defined. The phrase "incorporating equal ITV into the demographic parameters..." does not make sense to me, even after reading it several times. I would strongly urge the authors to revise the text again for clarity and perhaps to focus on simplifying the writing and making it more direct, with less jargon and undefined terminology. I suppose the main hurdle to overcome is explaining the existing theoretical framework satisfactorily.

My understanding of the theory is that whereas dominant species may occupy only a few ecological niches with abundant resources (certainly there are some dominant species that occupy regions of low resource availability that inferior species can't utilize. So I don't understand the focus on high resource availability), inferior species may use ITV as a means to capitalize on/occupy the remaining open niches. But isn't it also possible that inferior species are inferior because they have a limited ability to express variation in their traits (a low acclimation potential and low ITV?). The authors say that in a second scenario (when dominant species occupy most of the niches) that inferior species may be constrained to low ITV but I don't think you can assign causality in this way, for the reasons I described above. In short, a more nuanced explanation of the theory (with examples and a description of exceptions) would be particularly useful, and is what I'd recommended in the last revision. I understand that the authors weren't able to find empirical examples, but they could still put forth some examples even if there aren't published studies to go along with them. Maybe even a nice conceptual diagram would help. I understand that they are simply describing an existing body of work and not putting forth this theory for the first time, but if readers don't understand it then the empirical tests won't make sense either.

Going back to the response letter, you say that there is indirect evidence that ITV is greater in inferior species, yet you provide little detail here and don't explain the causal mechanisms. The example of *Potentilla* doesn't mention that this species is inferior. The following example is also for a clonal species, which is a bit odd, and then you only say that competition led to greater ITV but you still haven't explained how an inferior species would have higher ITV. Only one sentence in this rebuttal paragraph specifies the use of an inferior species, and that was for the *Quercus*. In short, I wasn't looking for more explanation about how ITV influences competition or vice versa, I was looking for clear examples of an inferior species having less ITV.

Another example in the response letter where I don't think you've provided a clear explanation (Line 405): "ITV promotes coexistence only when inferior species exhibit much higher ITV, because the greater ITV in inferior species causes some individuals with equal or lower competitive sensitivity versus superior species. This bolsters the realized population growth rate of inferior species and enables species coexistence." Greater ITV in all traits or traits that reflect resource use? Or ITV in life history traits? It seems like a big stretch to say greater ITV in general. And when you say much higher ITV, do you mean higher relative to superior species in the same environment or just generally speaking? Can you be more specific?

Specific comments:

Line 78: be careful with English grammar. I found several mistakes (Lines 46, 63, 67, 95). Here it should probably say "phenotypic plasticity resulting from competitive interactions can increase ITV, resulting in complex interactions between ITV and competition." And then here you really need an example: "For example, when the roots of neighboring plants come into contact with one another, one species may subsequently produce deeper roots in order to penetrate deeper into the soil and reduce competition. In other cases, both species may produce deeper roots at the same time, resulting in even stronger competition." The example you provided about the ramet and connectance traits is vague and therefore not particularly helpful. And I also don't know what a connectance trait is in the first place.

Line 86: in other words, there is asymmetry in the effect of competitive interactions for ITV and vice versa

In response to reviewer 1's comment and your reply: when you grew the plants in competition and then alone, was the latter specifically meant to identify the fundamental niche?

Line 95: I don't understand the explanation for why the inferior species would not adjust ITV. Above you said that it would and not you are saying it won't because there are other even more inferior species interacting with it.

Line 97: what do you mean by higher order interactions?

Line 103: I am still confused by the use of a homogeneous environment. If there's heterogeneity, it simply means that there could be high resource patches mixed with low resource patches, therefore, environmentally driven ITV (due to plasticity or local adaptation) is likely to have some influence on species interactions. In other words, the environment and presence of a competitor together influence the amount of ITV in the inferior species. When all patches are the same though, it should matter whether we are talking about a high resource or a low resource environment. If low resources, then there could be filtering of species as you say, but I don't see how a filter would necessarily restrict the range of means and also the amount of ITV. Can you comment further on how the relationship between ITV and competition would differ with resource availability (not heterogeneity)?

Line 104: across a gradient of resource availability or across spatially heterogeneous environments? Environmental conditions is vague.

Line 117: here you should just say what method you used. Whether it was looking at SD or CV.

Line 493 of response letter: 12-25 cm is a large amount of variation in seedling height. That's more than double at the initiation of the study. I'm glad you include the results from the mixed effects model but shouldn't initial plant size be included in all the other models if it had a significant impact on competition? This seems like a big deal.

Line 344 of main text: remove the word "slightly", these are acidic soils

Figure 2. are the orange lines in panels a and d the mean change when comparing inferior to superior?

Line 210: rather than just keep repeating the interspecific mean variance trade-off, you should use plain English to summarize. In other words, in a homogeneous environment with no competition, there was no evidence that inferior species (with lower mean trait values) also had higher ITV.

Line 234: same comment. Instead of competitive suppression just say what you mean in plain English rather than with lots of terms and jargon.

Line 251: grammar. Become not became.

Line 316: grammar. "of its importance" not on

Line 318: response surfaces

Reviewer #3 (Remarks to the Author):

The authors have address all of my concerns with the previous version, and the revised text is much clearer. I think this study will make a nice contribution to the field. I have only a few minor suggestions:

Line 28-32: Results should be reported in the past tense (e.g., No trade-off WAS found...)

Line 42: remove "previous unrecognized". It has been recognized for over a decade – see Clark 2010 Science and Clark et al. 2010 Ecol Monographs (you should look at these closely and see if you should cite them here or elsewhere since they seem highly relevant)

Line 67 – change 'rest' to 'remaining'

Line 68 – remove 'have to' or else rephrase because it is not grammatically correct

Line 84 – 'increases' should be 'increased'

Line 129 – 'greater increase', it is not clear an increase over what. I think you mean in a competitive or heterogenous environment compared to a competitive-free homogenous environment, but you should make that clear.

Line 135 – it will be unclear to readers what Phase I and II are since you have not yet explain it in the main text. I suggest adding a few words to make it clear what they mean.

Line 163 – add 'the' before 'two', otherwise it sounds like there were more than 2 deciduous species and you just chose to remove 2 of them

Line 331 – add 'potential' before 'importance'. As pointed out by another reviewer, you cannot prove its importance for coexistence with this study.

Line 547 – change "was" to "were"

Response letter to the reviewers' comments

Black: reviewer comments; Blue: our responses; Purple Italic: our revisions

Reviewer #2 (Remarks to the Author):

General comments

1. This manuscript seeks to clarify how intraspecific trait variation can influence competitive interactions in superior (dominant) versus inferior species, considering resource availability. I appreciate that the authors have included much more information about the methods, and have tried to clarify the introduction and discussion in several places. However, I am still finding this paper hard to follow, unfortunately. I understand the big picture but the explanation of the details still needs work. In reading the response to reviewer 1, I thought it was helpful to see a more formal definition of the mean-variance trade-off. However, I still don't think that the mechanistic explanation is clear enough for a broad audience that is unfamiliar with this work. Reading the main text (lines 52-63), it is still confusing, and written at a high level rather than being written at such a level that anyone could understand it. For example, the term competitive sensitivity is never defined, demographic rates are never defined. The phrase "incorporating equal ITV into the demographic parameters..." does not make sense to me, even after reading it several times. I would strongly urge the authors to revise the text again for clarity and perhaps to focus on simplifying the writing and making it more direct, with less jargon and undefined terminology. I suppose the main hurdle to overcome is explaining the existing theoretical framework satisfactorily.

Response: We sincerely appreciate your constructive feedback and encouragement regarding our most recent revision. Following your suggestions, we have streamlined this mechanistic explanation of the theory by eliminating unnecessary jargon and focusing directly on the essence of the matter (see lines 52-66). For further clarity, we have incorporated an example related to seed quality to better elucidate this theoretical framework for a broad audience.

Lines 52-66:

"Such is the case of tree species whose seed quality strongly correlates with their competitive ability at the seedling stage¹³, a high diversity of seed qualities can allow inferior species to produce some highly competitive individuals. However, any competitive advantage gained by an inferior species through such variation is outweighed if the superior species has similarly high ITV, and this outweighing occurs because the superior species has an inherently elevated mean seed quality. Equivalent ITV thus fails to provide the inferior species with a foothold over the superior species due to this baseline discrepancy in seed quality¹². The coexistence is promoted only when the inferior species exhibits greater trait variation among its conspecifics than the superior species¹⁰. This phenomenon is known as the interspecific mean-variance trade-off¹². When this trade-off occurs, some individuals of the inferior species have the same or greater trait value (i.e., seed quality) than do those of the superior species¹⁴, thus bolstering the persistence of their populations^{11,15}."

2. My understanding of the theory is that whereas dominant species may occupy only a few ecological niches with abundant resources (certainly there are some dominant species that occupy regions of low resource availability that inferior species can't utilize. So I don't understand the focus on high resource availability), inferior species may use ITV as a means to capitalize on/occupy the remaining open niches. But isn't it also possible that inferior species are inferior because they have a limited ability to express variation in their traits (a low acclimation potential and low ITV?). The authors say that in a second scenario (when dominant species occupy most of the niches) that inferior species may be constrained to low ITV but I don't think you can assign causality in this way, for the reasons I described above. In short, a more nuanced explanation of the theory (with examples and a description of exceptions) would be particularly useful, and is what I'd recommended in the last revision. I understand that the authors weren't able to find empirical examples, but they could still put forth some examples even if there aren't published studies to go along with them. Maybe even a nice conceptual diagram would help. I understand that they are simply describing an existing body of work and not putting forth this theory for the first time, but if readers don't understand it then the empirical tests won't make sense either.

Response: Thanks for your alternative explanation of the diminished ITV in inferior species. We have incorporated this compelling possibility into our revised exposition of the theoretical framework, including explicative examples and examinations of exceptions (see lines 72-78). Accordingly, we expanded our discussion on scenarios wherein inferior species exhibit low ITV (see lines 78-80). To further visualization these explanations, we have additionally constructed and included a new conceptual diagram (Fig. S1) in the Supplementary Information. We hope these changes and additions are helpful to illustrate the theoretical framework of our study.

Fig. S1 | Two distinct scenarios delineate the relative magnitude of intraspecific trait variability (ITV) in superior (green curve) and inferior (red curve) species. In scenario (a), the inferior species has greater ITV than the superior competitor. This could manifest, using root as an example, by the following mechanism – superior species occupy a few underground resource

niches while the inferior species actively adapt or passively exploit the remaining accessible resource niches. Occupying multiple niches culminates in a multimodal trait distribution, translating to amplified ITV for root depth in the inferior species. Conversely, scenario (b) depicts the inferior species possessing less ITV than the superior counterpart. Contributing factors may be inherent constraints in trait plasticity or adaptability that contribute to their competitive disadvantages. The colored solid lines depict trait distribution examples for both inferior and superior species under two scenarios.

Lines 72-78:

“For example, while superior species may benefit from deep roots to monopolize soil resources, inferior species might adapt either by extending their roots deeper²² or by spreading them laterally to access shallow soil resources²³, resulting in a multipeaked root depth trait distribution with higher ITV (Fig. S1a). Conversely, if the superior species already occupies most of the niche space^{24,25}, the inferior species may compress its trait values to maximize its survival in the remaining limited niche space. This may result in a smaller ITV in the inferior species.”

Lines 78-80:

“Alternatively, inferior species might inherently face limitations in expressing trait variability^{26,27}, which would lead to low ITV (Fig. S1b).”

3. Going back to the response letter, you say that there is indirect evidence that ITV is greater in inferior species, yet you provide little detail here and don't explain the causal mechanisms. The example of *Potentilla* doesn't mention that this species is inferior. The following example is also for a clonal species, which is a bit odd, and then you only say that competition led to greater ITV but you still haven't explained how an inferior species would have higher ITV. Only one sentence in this rebuttal paragraph specifies the use of an inferior species, and that was for the *Quercus*. In short, I wasn't looking for more explanation about how ITV influences competition or vice versa, I was looking for clear examples of an inferior species having less ITV.

Response: Thank you for specifying the information that would be most pertinent. We found an example in Carmona et al. (2019) demonstrating competitively inferior species displayed higher ITV than superior species. Compared to traits measured in absence of competition, the traits of the inferior species became more similar to those of the superior when the two species grew in pairwise competition. As a result, competitive hierarchies were reduced, favoring the persistence of the competitively inferior species. This example has been incorporated into the revised text (see lines 99-101), replacing the previous instance involving clonal plants.

Additionally, in response to your request for an example of an inferior species with less ITV, we referred to Ashton et al. (2010). Specifically, despite a superior competitor exploiting increased availability of certain nitrogen forms through phenotypic plasticity, the inferior species did not show a parallel shift in its nitrogen absorption profile when grown in competition.

Lines 99-101:

“Similarly, in an experiment with Mediterranean annual plant species, the inferior species experienced greater changes in trait values when competing with superior species, resulting in reduced competitive hierarchies and favoring coexistence⁷.”

4. Another example in the response letter where I don't think you've provided a clear explanation (Line 405): “ITV promotes coexistence only when inferior species exhibit much higher ITV, because the greater ITV in inferior species causes some individuals with equal or lower competitive sensitivity versus superior species. This bolsters the realized population growth rate of inferior species and enables species coexistence.” Greater ITV in all traits or traits that reflect resource use? Or ITV in life history traits? It seems like a big stretch to say greater ITV in general. And when you say much higher ITV, do you mean higher relative to superior species in the same environment or just generally speaking? Can you be more specific?

Response: We apologize for the lack of clarity in our previous response. While the concept of "greater ITV" is highly generalizable across contexts and operationalized differently per study (e.g. Hart et al. 2016; Lichstein et al. 2007), herein we refer specifically to greater variability in traits directly linked to competitive ability (e.g. seed quality) within inferior species versus superior species given similar environmental conditions. For enhanced precision, we have made the term more specific in our manuscript at lines 53-61 and 63-66.

Lines 53-61:

“Such is the case of tree species whose seed quality strongly correlates with their competitive ability at the seedling stage¹³, a high diversity of seed qualities can allow inferior species to produce some highly competitive individuals. However, any competitive advantage gained by an inferior species through such variation is outweighed if the superior species has similarly high ITV, and this outweighing occurs because the superior species has an inherently elevated mean seed quality. Equivalent ITV thus fails to provide the inferior species with a foothold over the superior species due to this baseline discrepancy in seed quality¹².”

Lines 63-66:

“When this trade-off occurs, some individuals of the inferior species have the same or greater trait value (i.e., seed quality) than do those of the superior species¹⁴, thus bolstering the persistence of their populations^{11,15}.”

Specific comments:

5. Line 78: be careful with English grammar. I found several mistakes (Lines 46, 63, 67, 95). Here it should probably say “phenotypic plasticity resulting from competitive interactions can increase ITV, resulting in complex interactions between ITV and competition.” And then here you really need an example: “For example, when the roots of neighboring plants come into contact with one another, one species may subsequently produce deeper roots in order to penetrate deeper into the soil and reduce competition. In other cases, both species may produce deeper roots at the same time, resulting in even stronger competition.” The example you provided about the ramet and

connectance traits is vague and therefore not particularly helpful. And I also don't know what a connectance trait is in the first place.

Response: Thank you for your constructive feedback. We have meticulously revised the manuscript to address all grammatical errors. Additionally, we have enlisted the Premium Editing services of American Journal Experts (AJE), with the verification code 9B3D-052B-155A-EFCE-C97P, aiming to refine the manuscript's clarity and appeal to diverse audiences.

Moreover, in response to your suggestion, we have incorporated an example explaining how root phenotypic plasticity and associated influences shape ITV into the manuscript (lines 89-92).

Lines 89-92:

“For example, when the roots of neighboring plants come into contact, one species may respond by producing deeper roots, mitigating competition²². Alternatively, both species may simultaneously exhibit root elongation, intensifying competition³³.”

6. Line 86: in other words, there is asymmetry in the effect of competitive interactions for ITV and vice versa.

Response: Yes, we have clarified and emphasized this point in lines 101-103.

Lines 101-103:

“These findings imply that the impact of competition on ITV might be asymmetrical between superior and inferior species³⁴, triggering the interspecific mean-variance trade-off.”

7. In response to reviewer 1's comment and your reply: when you grew the plants in competition and then alone, was the latter specifically meant to identify the fundamental niche?

Response: When allowed to grow alone, the abiotic conditions of seedlings did approximate their fundamental niche (Ashton et al. 2010). However, given the constrained type and range of environmental variables in our study, alone growth conditions likely did not encapsulate the full spectrum of abiotic environments permissible to each species' survival without competition. Our primary objective in utilizing alone and mixed-species plantings was to investigate how competition shapes ITV.

8. Line 95: I don't understand the explanation for why the inferior species would not adjust ITV. Above you said that it would and not you are saying it won't because there are other even more inferior species interacting with it.

Response: Apologies for any confusion. We recognize that our discussion regarding potential adjustments of ITV in inferior species under multispecies interactions lacked clarity. Earlier in the manuscript, we looked at how inferior species can change their ITV in response to pairwise competition. Then we discussed that inferior species may respond differently under more complex multispecies interactions for two key reasons: 1) Ecological niche saturation: In competitive environments with higher species number, ecological niches become widely occupied, resulting in

fewer unoccupied niche spaces. This occupancy saturation limits opportunities for inferior species to effectively utilize ITV adjustments to access open niche dimensions. 2) Complex competitive dynamics: Multispecies systems elicit more complex interactions between species, similar to “rock-paper-scissors” cycles. Although an inferior species could alter traits to gain an advantage over one competitor, such ITV shifts may inadvertently handicap it against another species. These complex interactions constrain the degree to which species can substantially modify traits conferring multilateral competitors. We have revised the relevant text in lines 110-119 to accurately reflect this feedback.

Lines 110-119:

“Unlike simple pairwise competition, multispecies interactions involve more than two species, thus possibly reducing unoccupied niche space as the total number of species increases³⁷. However, heterogeneous abiotic environments can expand the overall available niche space, potentially increasing unoccupied niches^{36,38}. The uncertainty surrounding vacant niches in such ecosystems makes it difficult to predict how inferior species might adapt their ITV to access untapped resources and respond to competition. Furthermore, abiotic environments may also affect the magnitude of ITV by either directly filtering out individuals with unfit trait values^{20,39} or indirectly adjusting interspecific competition, which in turn alters the extent and direction of plastic responses of traits⁴⁰.”

9. Line 97: what do you mean by higher order interactions?

Response: Higher-order interactions are ecological scenarios where a species' direct impact on another is influenced by the presence of additional species, an occurrence primarily noted in multi-species systems (Levine et al. 2017). This dynamic unfolds as one species undergoes trait plasticity changes in response to another, which in turn modifies its direct effects on a third party. Consequently, these are often termed as trait-mediated indirect interactions (Padilla et al. 2013). While these interactions may have unforeseen effects on ITV, their inherent complexity makes them challenging to thoroughly investigate in this context. Therefore, to maintain focus and clarity, we've removed related content from the Introduction section.

10. Line 103: I am still confused by the use of a homogeneous environment. If there's heterogeneity, it simply means that there could be high resource patches mixed with low resource patches, therefore, environmentally driven ITV (due to plasticity or local adaptation) is likely to have some influence on species interactions. In other words, the environment and presence of a competitor together influence the amount of ITV in the inferior species. When all patches are the same though, it should matter whether we are talking about a high resource or a low resource environment. If low resources, then there could be filtering of species as you say, but I don't see how a filter would necessarily restrict the range of means and also the amount of ITV. Can you comment further on how the relationship between ITV and competition would differ with resource availability (not heterogeneity)?

Response: Thank you for highlighting the lack of clarity. Here is the explanation about the filtering process of low resource environments on species' trait distributions (Fig. S8): With ITV, adaptations towards the environment vary across individuals. When no individuals demonstrate suitable adaptations, the entire species faces elimination (species 4 with green trait distribution). However, if only certain individuals are maladapted, the filter selectively removes those, leaving adapted members (species 1 with red trait distribution). Consequently, the surviving population experiences tightened ITV around a shifted mean trait value aligned with the selective pressure. We have added this conceptual diagram in Discussion (lines 235-240).

Resource availability can modulate the association between ITV and competition. For instance, in high-resource settings (e.g., environment block 1 with abundant light and water), inferior species pressures exhibited higher ITV (slope of -8.84 in Figs. 1e and S5b). Under resource-constrained environments (e.g., block 4 with scarce light and nutrients), inferior species still exhibited elevated ITV but the relationship was substantially attenuated (slope of -2.58 in Fig. S5b). This aligns with the stress gradient hypothesis, whereby species in resource-limited environments experience restricted growth and dampened competition (Stein et al. 2014). While we did not exhaustively investigate these nuanced relationships, we document these patterns here (Discussion, lines 284-293) as meriting detailed exploration in future work.

Fig. S8 | A conceptual diagram delineates abiotic environmental filtering operating at the individual level across various species (colored trait distribution curves). If all individuals of a species are maladapted, the species go extinction (species 4 with green trait distribution curve). Alternatively, with variable ITV, if a subset of individuals are maladapted, the filter eliminates only these individuals, enabling the persistence of adapted subset (species 1 with red curve). The resulting truncated population passing through this bottleneck exhibits reduced ITV and a changed mean trait value. Essentially, environmental filtering alters trait distribution via this ITV reduction and mean shift.

Lines 235-240:

“In a competition-free homogeneous environment, all individuals, regardless of the status of the species in the competitive hierarchy, are subject to similar abiotic selection and use the same set of resources (e.g., Fig. S8). This results in similar variation in most traits between inferior and superior species and lower levels of ITV in a competition-free homogeneous environment than in heterogeneous or competitive environments⁴⁷.”

Lines 284-293:

“Moreover, resource availability can also modulate the association between ITV and competition. For instance, in environments with abundant resources (e.g., environment block 1 with abundant light and water), inferior species exhibited higher ITVs (slope of -8.84 in Figs. 1e and S5b), while in resource-constrained conditions (e.g., block 4 with scarce light and nutrients), this relationship weakened (slope of -2.58 in Fig. S5b). This finding aligns with the stress gradient hypothesis³⁶, whereby species in resource-limited environments experience restricted growth and dampened competition. These findings highlight the need for further exploration of the effect of resource availability on the relationship between ITV and competition.”

11. Line 104: across a gradient of resource availability or across spatially heterogeneous environments? Environmental conditions is vague.

Response: It is whether the trade-off still hold across spatially heterogeneous environments. Corrected.

12. Line 117: here you should just say what method you used. Whether it was looking at SD or CV.

Response: We employed the CV statistic to assess ITV for a single trait, specifically using *Bao's CV*. This method has been validated for a more accurate estimation of ITV (Yang *et al.* 2020). We have now incorporated this information into the main text for clarity (see line 135-137).

Lines 135-137:

*“We correspondingly estimated the ITVs for each species in both treatments in each scenario using two methods: multidimensional trait space⁴¹⁻⁴³ and individual trait variability (*Bao's CV*)^{6,7,44,45}.”*

13. Line 493 of response letter: 12-25 cm is a large amount of variation in seedling height. That's more than double at the initiation of the study. I'm glad you include the results from the mixed effects model but shouldn't initial plant size be included in all the other models if it had a significant impact on competition? This seems like a big deal.

Response: Thank you for your suggestion. Please note that the 12-25 cm differences in initial plant height largely stem from interspecific distinctions (see Table S7). These interspecific variances reflect intrinsic differences in life stage development across species and are difficult to control in an experiment. Rather, intraspecific differences in initial height were carefully controlled for by

selecting similarly sized conspecifics during transplanting (line 426-429 and 489-490). As such, intraspecific height variations are relatively minor compared to interspecific discrepancies.

Additionally, we conducted supplementary analyses to evaluate potential impacts of initial plant height on the relationship between competition and ITV. Specifically, we included the average initial height of each species as a fixed effect in the mixed-effects model. The new results indicated that initial plant height did not exert a significant effect on the ITV of the species, both in homogeneous ($P = 0.936$) and heterogeneous ($P = 0.158$) environments (refer to Table R1). Importantly, the new results (Table R1) were consistent with our previous findings (Fig. 2 and Table S1), indicating that initial size differences between seedlings do not skew our detected conclusions. In fact, our previous model already account for the interspecific differences by including species identity as a random factor. This may explain the consistent outputs between models with and without initial plant sizes. Given the non-significant impact of initial height on our model's results, we opted to exclude this parameter from the final model in order to retain parsimony and emphasis on the variables of primary interest.

Table R1 Summary of the mixed-effects models employed to explore the relationships involving competitive treatments (competition-free and competition, abbreviated as ‘CompTreat’), competitiveness (RII), their interaction, and average initial height of each species concerning species’ ITV within the context of multispecies experiment. These analyses were separately conducted in both homogeneous (specifically, environmental block 1, which corresponds to the Phase I environment) and heterogeneous environments, with species as a random factor. Values in bold indicate statistically significant effects ($P < 0.05$).

	Predictors	Estimates	CI	P
Multispecies experiment in a homogeneous environment	CompTreat_competition-free	0.93	-5.40 – 7.26	0.738
	CompTreat_competition	0.53	-5.80 – 6.86	0.850
	competitiveness	-0.51	-8.14 – 7.11	0.878
	initial height	-0.01	-0.21 – 0.20	0.936
	CompTreat_competition* competitiveness	-7.14	-13.97 – -0.32	0.043
			Pseudo R^2	0.856
Multispecies experiment in a heterogeneous environment	CompTreat_competition-free	-5.96	-15.24 – 3.32	0.173
	CompTreat_competition	-5.54	-14.83 – 3.74	0.201
	competitiveness	-15.26	-26.42 – -4.09	0.014
	initial height	0.20	-0.10 – 0.50	0.158
	CompTreat_competition* competitiveness	-6.26	-12.13 – -0.39	0.040

Lines 426-429:

“To ensure a fair comparison between treatments, the initial height and base diameter of the seedlings were kept similar for each species among the pots. If a seedling died within the first month after transplanting, it was replaced with a similar-sized conspecific seedling.”

Lines 489-490:

“Healthy seedlings of similar initial heights were selected for each species (Table S7) and transplanted into the pots.”

14. Line 344 of main text: remove the word “slightly”, these are acidic soils

Response: We have removed the word "slightly" from manuscript.

15. Figure 2. are the orange lines in panels a and d the mean change when comparing inferior to superior?

Response: In panels a and d in Fig. 2, the orange dots represent the mean of ITV for eight superior and inferior species. The orange line segment connecting these dots indeed illustrates the difference in ITV between competing species. We have updated the figure notes (lines 889-892) to express this more explicitly.

Lines 889-892:

“The orange dots in a and d represent the mean ITVs of the eight superior (or inferior) species. The grey and orange line connecting these dots visually depicts the difference in ITV between competing species, and significance was tested by paired Wilcoxon rank-sum test.”

16. Line 210: rather than just keep repeating the interspecific mean variance trade-off, you should use plain English to summarize. In other words, in a homogeneous environment with no competition, there was no evidence that inferior species (with lower mean trait values) also had higher ITV.

Response: We have revised the pertinent sections (lines 230-233) to state more directly.

Lines 230-233:

“Under a competition-free homogeneous environment, our observations did not support the hypothesis that species with lower average competitive abilities exhibit higher levels of ITV, countering the assumption of an interspecific mean-variance trade-off.”

17. Line 234: same comment. Instead of competitive suppression just say what you mean in plain English rather than with lots of terms and jargon.

Response: Thanks to your constructive comment. We have replaced the term *"intensity of competition (RII), as evidenced by a relative decrease in overall biomass."* (lines 258-259).

18. Line 251: grammar. Become not became.

Response: Corrected.

19. Line 316: grammar. “of its importance” not on

Response: Corrected.

20. Line 318: response surfaces

Response: Corrected.

Reviewer #3 (Remarks to the Author):

The authors have address all of my concerns with the previous version, and the revised text is much clearer. I think this study will make a nice contribution to the field. I have only a few minor suggestions:

1. Line 28-32: Results should be reported in the past tense (e.g., No trade-off WAS found...)

Response: revised.

2. Line 42: remove “previous unrecognized”. It has been recognized for over a decade – see Clark 2010 Science and Clark et al. 2010 Ecol Monographs (you should look at these closely and see if you should cite them here or elsewhere since they seem highly relevant)

Response: Thanks for your valuable suggestion. After a thorough review of the mentioned papers, we have rephrased the sentence, omitting “previous unrecognized”. Additionally, we have accordingly cited both references in the revised manuscript (line 43).

3. Line 67 – change ‘rest’ to ‘remaining’

Response: Done.

4. Line 68 – remove ‘have to’ or else rephrase because it is not grammatically correct

Response: Thanks. We have revised the sentence (lines 75-78) for clarity and correctness.

Lines 75-78:

“Conversely, if the superior species already occupies most of the niche space^{24,25}, the inferior species may compress its trait values to maximize its survival in the remaining limited niche space. This may result in a smaller ITV in the inferior species.”

5. Line 84 – ‘increases’ should be ‘increased’

Response: Done.

6. Line 129 – ‘greater increase’, it is not clear an increase over what. I think you mean in a competitive or heterogenous environment compared to a competitive-free homogenous environment, but you should make that clear.

Response: Thanks. To enhance clarity, we have rephrased the sentence (lines 149-152).

Lines 149-152:

“We found that the emergence of the trade-off was driven by a greater increase in ITV in inferior species than in superior species when competing species transitioned from competition-free homogeneous environments to competitive or heterogeneous environments.”

7. Line 135 – it will be unclear to readers what Phase I and II are since you have not yet explain it in the main text. I suggest adding a few words to make it clear what they mean.

Response: Thank you for your insightful suggestion. We have incorporated a concise definition of Phase I and Phase II at the end of introduction section (lines 127, 130 and 132).

8. Line 163 – add ‘the’ before ‘two’, otherwise it sounds like there were more than 2 deciduous species and you just chose to remove 2 of them

Response: Great point! We have added “the” before “two”.

9. Line 331 – add ‘potential’ before ‘importance’. As pointed out by another reviewer, you cannot prove its importance for coexistence with this study.

Response: Thanks for your valuable feedback. We have rephrased this sentence (line 366-367).

Lines 366-367:

“These findings provide the first experimental evidence for this trade-off, providing empirical evidence for its potential importance in species coexistence.”

10. Line 547 – change “was” to “were”

Response: Done.

Reference

- Ashton, I.W., Miller, A.E., Bowman, W.D. & Suding, K.N. (2010). Niche complementarity due to plasticity in resource use: plant partitioning of chemical N forms. *Ecology*, 91, 3252–3260.
- Carmona, C.P., de Bello, F., Azcárate, F.M., Mason, N.W.H. & Peco, B. (2019). Trait hierarchies and intraspecific variability drive competitive interactions in Mediterranean annual plants. *Journal of Ecology*, 107, 2078–2089.
- Levine, J.M., Bascompte, J., Adler, P.B. & Allesina, S. (2017). Beyond pairwise mechanisms of species coexistence in complex communities. *Nature*, 546, 56–64.
- Padilla, F.M., Mommer, L., Caluwe, H. de, Smit-Tiekstra, A.E., Wagemaker, C.A.M., Ouborg, N.J., et al. (2013). Early Root Overproduction Not Triggered by Nutrients Decisive for Competitive Success Belowground. *Plos one*, 8, e55805.
- Stein, A., Gerstner, K. & Kreft, H. (2014). Environmental heterogeneity as a universal driver of species richness across taxa, biomes and spatial scales. *Ecology Letters*, 17, 866–880.
- Yang, J., Lu, J., Chen, Y., Yan, E., Hu, J., Wang, X., et al. (2020). large underestimation of intraspecific trait variation and its improvements. *Frontiers in Plant Science*, 11, 53.
- Yang, J., Lu, J., Wang, R., Wang, X., Li, S. & Shen, G. (2021). Importance and benefit of incorporating the responses of species mean trait values in trait-based community assembly. *Ecological Indicators*, 130, 108095.